# The Parcels v2.0 Lagrangian framework: new field interpolation schemes

Philippe Delandmeter[1] and Erik van Sebille[1]

[1]Utrecht University, Institute for Marine and Atmospheric Research, Princetonplein 5, 3584 CC Utrecht, The Netherlands

**Correspondence:** Philippe Delandmeter (p.b.delandmeter@uu.nl)

**Abstract.** With the increasing amount of data produced by numerical ocean models, so increases the need for efficient tools to analyse these data. One of these tools is Lagrangian ocean analysis, where a set of virtual particles are released and their dynamics is integrated in time based on fields defining the ocean state, including the hydrodynamics and biogeochemistry if available. This popular methodology needs to adapt to the large variety of models producing these fields at different formats.

This is precisely the aim of Parcels, a Lagrangian ocean analysis framework designed to combine (1) a wide flexibility to model particles of different natures and (2) an efficient implementation in accordance with modern computing infrastructure. In the new Parcels v2.0, we implement a set of interpolation schemes to read various types of discretised fields, from rectilinear to curvilinear grids in the horizontal direction, from $z$- to $s$- levels in the vertical and using grid staggering with the Arakawa's A-, B- and C- grids. In particular, we develop a new interpolation scheme for a three-dimensional curvilinear C-grid and analyse its properties.

Parcels v2.0 capabilities, including a suite of meta-field objects, are then illustrated in a brief study of the distribution of floating microplastic in the North West European continental shelf and its sensitivity to various physical processes.

## 1  Introduction

Numerical ocean modelling has evolved tremendously in the past decades, producing more accurate results, with finer spatial and time resolutions (Prodhomme et al., 2016). With the accumulation of very large data sets resulting from these simulations, the challenge of ocean analysis has grown. Lagrangian modelling is a powerful tool to analyse flows in several fields of engineering and physics, including geophysics and oceanography (van Sebille et al., 2018).

While Lagrangian modelling can be used to simulate the flow dynamics itself (e.g. Monaghan, 2005), most of the modelling effort in geophysical fluid dynamics is achieved with an Eulerian approach. Lagrangian methods can, in turn, be used to analyse the ocean dynamic given the flow field from an Eulerian model. The flow field can also be taken from land-based measurement such as high frequency radar (Rubio et al., 2017), and satellite imagery that measure altimetry (Holloway, 1986) or directly the currents using Doppler radar (Ardhuin et al., 2018).

Lagrangian analysis simulates the pathways of virtual particles, that can represent water masses, tracers such as temperature, salinity or nutrients, or particulates like sea grass (e.g. Grech et al., 2016), kelp (e.g. Fraser et al., 2018), coral larvae (e.g. Thomas et al., 2014), plastics (e.g. Lebreton et al., 2012; Onink et al., 2019), fish (e.g Phillips et al., 2018), icebergs (e.g. Marsh

et al., 2015), etc. It is used at a wide range of time and spatial scales as for example the modelling of plastic dispersion, from coastal applications (e.g. Critchell and Lambrechts, 2016) to regional (e.g. Kubota, 1994) or global scales (e.g. Maximenko et al., 2012).

The method consists in advancing, for each particle, the coordinates and other state variables by first interpolating fields of interest, as the velocity or any tracer, at the particle position and integrating in time the ordinary differential equations defining the particle dynamics.

A number of tools are available to track virtual particles, with diverse characteristics, strengths and limitations including Ariane (Blanke and Raynaud, 1997), TRACMASS (Döös et al., 2017), CMS (Paris et al., 2013) and OpenDrift (Dagestad et al., 2018). An extensive list and description of Lagrangian analysis tools is provided in van Sebille et al. (2018). One of the tools is Parcels.

Parcels ("Probably A Really Computationally Efficient Lagrangian Simulator") is a framework for computing Lagrangian particle trajectories (http://www.oceanparcels.org, Lange and van Sebille, 2017). The main goal of Parcels is to process the continuously increasing amount of data generated by the contemporary and future generations of ocean general circulation models (OGCMs). This requires two important features of the model: (1) not to be dependent on one single format of fields and (2) to be able to scale up efficiently to cope with up to petabytes of external data produced by OGCMs. In Lange and van Sebille (2017), the concept of the model was described and the fundamentals of Parcels v0.9 were stated. Since this version, the model essence has remained the same, but many features were added or improved, leading to the current version 2.0. Among all the developments, our research has mainly focused to develop and implement into Parcels interpolation schemes to provide the possibility to use a set of fields discretised on various types of grids, from rectilinear to curvilinear in the horizontal direction, with $z$- or $s$- levels in the vertical, and using grid staggering with A-, B- and C- Arakawa staggered grids (Arakawa and Lamb, 1977). In particular an interpolation scheme for curvilinear C-grids, that was not defined in other Lagrangian analysis models, was developed for both $z$ and $s$-levels.

In this paper, we detail the interpolation schemes implemented into Parcels, with a special care in the description of the new curvilinear C-grid interpolator. We describe the new meta-field objects available into Parcels for easier and faster simulations. We prove some fundamental properties of the interpolation schemes. We then validate the new developments through a study of the sensitivity of floating microplastic dispersion and 3D passive particles distribution on the North West European continental shelf and discuss the results.

## 2    Parcels v2.0 development

To simulate particle transport in a large variety of applications, Parcels relies on two key features: (1) interpolation schemes to read external data sets provided on different formats and (2) customisable kernels to define the particle dynamics.

The interpolation schemes are necessary to obtain the field value at the particle 3D location. They have been vastly improved in this latest version 2.0. Section 2.1 describes the interpolation of fields and Section 2.2.1 their structure in Parcels.

The kernels are already available since the earliest version of Parcels (Lange and van Sebille, 2017). They are the core of the model, integrating particle position and state variables. The built-in kernels and the development achieved are briefly discussed in Section 2.2.2.

## 2.1 Field Interpolation

External data sets are provided to Parcels as a set of fields. Each field is discretised on a structured grid that provides the node locations and instants at which the field values are given. It is noteworthy that the fields in a field set are not necessarily based on the same grid. In the horizontal plane, rectilinear (Fig. 1a) and curvilinear (Fig. 1b) grids are implemented. Three-dimensional data are built as the vertical extrusion of the horizontal grid, either using $z$-levels (Fig. 1c) or $s$-levels (Fig. 1d). A three-dimensional mesh is a combination of a rectilinear or a curvilinear in the horizontal direction mesh with either $z$- or

$s$- levels in the vertical. Another class of meshes are the unstructured grids (e.g. Lambrechts et al., 2008), which are not yet supported in Parcels.

    Fields can be independent from each other (e.g. water velocity from one data set and wind stress from a different data set) and interpolated separately. Often, fields come from a same data set, for example when they result from a numerical model, and form a coherent structure, that must be preserved in Parcels; an example is the zonal and meridional components of the

velocity field. A coherent field structure is discretised on the same grid, but the variables are not necessarily distributed evenly, leading to a so-called staggered grid (Arakawa and Lamb, 1977). The various types of staggered grids have their pros and cons (Cushman-Roisin and Beckers, 2011). Parcels reads the more popular staggered grids occurring in geophysical fluid dynamics: A-, B- and C- grids. While A- and C- are fundamentally different (Fig. 2), the B-grid can be considered as a hybrid configuration in between A- and C- grids. D- and E- grids are less common and are not implemented into the framework so

far. The methods to interpolate fields on the A-, C- and B- grids are presented in this section.

### 2.1.1 A-grid

The A-grid is the un-staggered Arakawa's grid: zonal velocity ($u$), meridional velocity ($v$) and tracers ($T$) are collocated (Fig. 2a). This grid is used in many reanalysis data sets for global currents (e.g. Globcurrent, Rio et al., 2014) or tidal dynamics (e.g. TPXO, Egbert and Erofeeva, 2002), as well as regridded products such as OFES (Sasaki et al., 2008) or data sets on platforms

like the Copernicus Marine Environment Monitoring Service. In this section we consider first the two-dimensional case before describing three-dimensional fields.

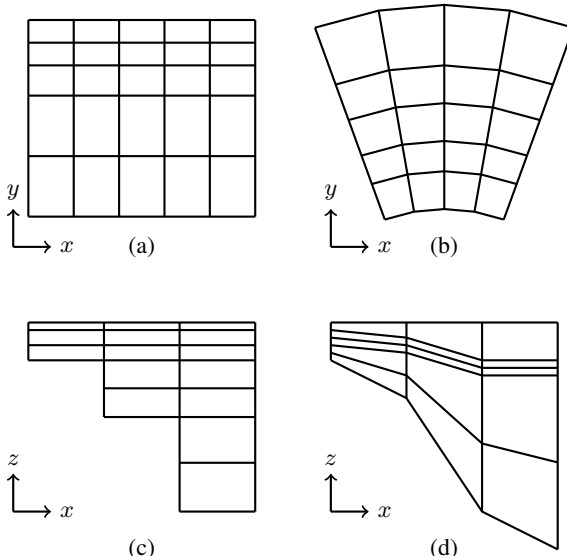

**Figure 1.** Grid discretisations processed by Parcels. In the horizontal plane: (a) rectilinear, (b) curvilinear; in the vertical: (c) $z$-levels and (d) $s$-levels. Four combinations of horizontal and vertical grids are possible to form a three-dimensional mesh.

### 2D field

In a two-dimensional context, field $f$ is interpolated in cell $(j,i)$ where the particle is located, with the bi-linear Lagrange polynomials $\phi_n^{2D}$ and the four nodal values $F_n$ with $n = 0,...,3$, surrounding the cell, resulting in the following expression:

$$f(x,y) = \sum_{n=0}^{3} \phi_n^{2D}(\xi,\eta)\, F_n, \tag{1}$$

$$\text{with } \xi,\eta \text{ s.t.} \begin{cases} x = \sum_n \phi_n^{2D}(\xi,\eta)\, X_n \\ y = \sum_n \phi_n^{2D}(\xi,\eta)\, Y_n. \end{cases} \tag{2}$$

$\xi$, $\eta$ are the relative coordinates in the unit cell (Fig. 3b), corresponding to the particle relative position in the physical cell (Fig. 3a). $(X_n, Y_n)$ are the coordinates of the cell vertices. The two-dimensional Lagrange polynomials $\phi_n^{2D}$ are the bi-linear functions:

$$\phi_0^{2D}(\xi,\eta) = (1-\xi)(1-\eta), \qquad \phi_1^{2D}(\xi,\eta) = \xi\,(1-\eta),$$
$$\phi_2^{2D}(\xi,\eta) = \xi\,\eta, \qquad \phi_3^{2D}(\xi,\eta) = (1-\xi)\,\eta.$$

Note that in a rectilinear mesh, solving Eq. 2 reduces to the usual solutions:

$$\xi = \frac{x - X_0}{X_1 - X_0}, \qquad \eta = \frac{y - Y_0}{Y_3 - Y_0}. \tag{3}$$

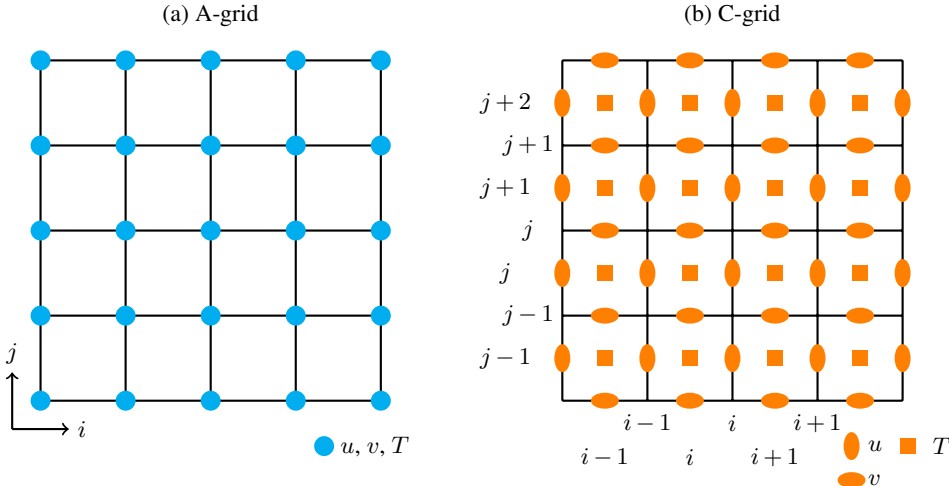

**Figure 2.** Arakawa's staggered grids (Arakawa and Lamb, 1977): (a) A-grid and (b) C-grid. In C-grid, $i$ and $j$ represent the variable column and row indexing in arrays where the variables are stored. The indexing of the C-grid follows the NEMO notations (Madec et al., 2016).

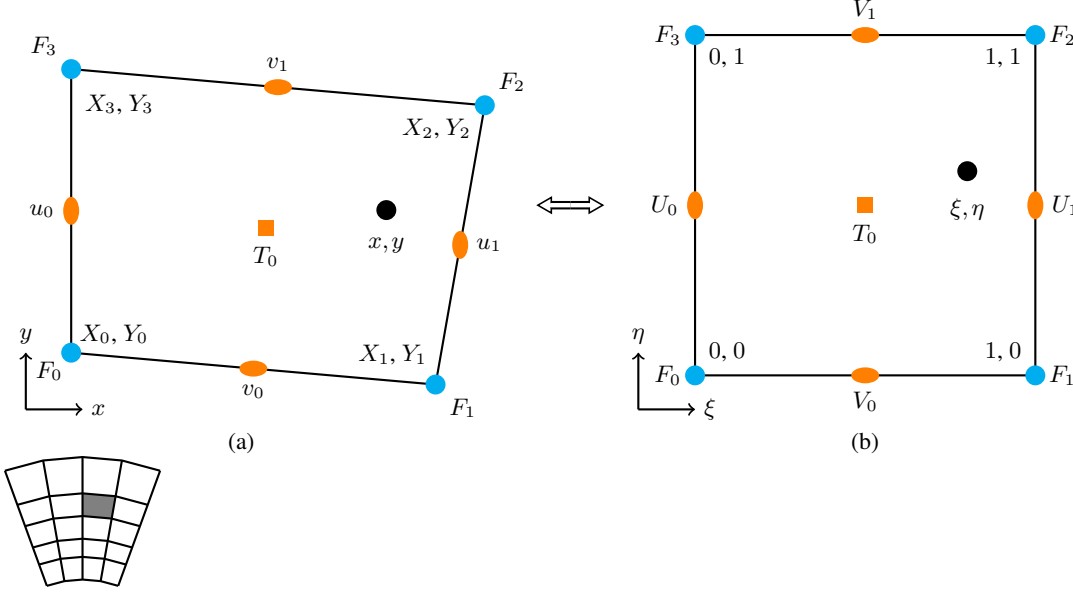

**Figure 3.** Positioning of the variables for a A-grid (blue nodes) and C-grid (orange nodes) cell with (a) physical coordinates in the mesh cell and (b) relative coordinates in the unit cell.

**3D field**

To read three-dimensional fields, for both $z$- and $s$- levels, the scheme interpolates the eight nodal values located on the hexahedron vertices using the tri-linear Lagrangian polynomials $\phi_n^{3D}$:

$$\phi_0^{3D}(\xi,\eta,\zeta) = (1-\xi)(1-\eta)(1-\zeta), \qquad \phi_1^{3D}(\xi,\eta,\zeta) = \xi\,(1-\eta)(1-\zeta),$$
$$\phi_2^{3D}(\xi,\eta,\zeta) = \xi\,\eta\,(1-\zeta), \qquad \phi_3^{3D}(\xi,\eta,\zeta) = (1-\xi)\,\eta\,(1-\zeta),$$
$$\phi_4^{3D}(\xi,\eta,\zeta) = (1-\xi)(1-\eta)\,\zeta, \qquad \phi_5^{3D}(\xi,\eta,\zeta) = \xi\,(1-\eta)\,\zeta,$$
$$\phi_6^{3D}(\xi,\eta,\zeta) = \xi\,\eta\,\zeta, \qquad \phi_7^{3D}(\xi,\eta,\zeta) = (1-\xi)\,\eta\,\zeta.$$

$\xi$ and $\eta$ are obtained as in the 2D case, using the first four vertices of the hexahedron. The vertical relative coordinate is obtained as:

$$\zeta = \frac{z - z_0}{z_1 - z_0}, \tag{4}$$

with

$$z_0 = \sum_{n=0}^{3} \phi_n^{2D} Z_n, \qquad z_1 = \sum_{n=4}^{7} \phi_n^{2D} Z_n. \tag{5}$$

The interpolation results in:

$$f(x,y,z) = \sum_{n=0}^{7} \phi_n^{3D}(\xi,\eta,\zeta)\, F_n. \tag{6}$$

### 2.1.2 C-grid

In a C-grid discretisation, the velocities are located on the cell edges and the tracers are at the middle of the cell (Fig. 2b). One could attempt to interpolate zonal and meridional velocities as if they would be on two different A-grids, shifted from half a cell, but this would not be in accordance with the C-grid formulation: in particular, such interpolation would break impermeability at the coastal boundary conditions. Instead, we use the interpolation principle based on the scheme used in the Lagrangian model TRACMASS (Jönsson et al., 2015; Döös et al., 2017), but generalised to curvilinear meshes.

The tracer is computed as a constant value all over the cell, in accordance with the mass conservation schemes of C-grids. The formulations for the two-dimensional and three-dimensional velocities consist of four steps:

1. define a mapping between the physical cell and a unit cell (as for the A-grid, Fig. 3);

2. compute the fluxes on the unit cell interfaces, as a function of the velocities on the physical cell interfaces;

3. interpolate those fluxes to obtain the relative velocity;

4. transform the relative velocity to the physical velocity.

Step 1 consists in applying Eq. 2 for the 2D case and Eqs. 2 and 4 for 3D fields. The other three steps are defined below, for both 2D and 3D fields.

**2D field**

The velocities at the edges of cell $(j, i)$ are $(u_{j+1,i}, u_{j+1,i+1}, v_{j,i+1}, v_{j+1,i+1})$ (Fig. 2b). This indexing was chosen to be consistent with the notation used by the NEMO model (Madec et al., 2016). For readability, they will be renamed in this section to $(u_0, u_1, v_0, v_1)$, using local indices (Fig. 3a) instead of the global indices . The velocity at $(x, y)$ is not obtained by linear interpolation but, like in finite-volume schemes, it is approximated by linearly interpolating the fluxes $(U_0, U_1, V_0, V_1)$ through the cell edges (Fig. 3b), that read (Step 2):

$$
\begin{cases}
U_0 = L_{03}\, u_0, \\
U_1 = L_{12}\, u_1, \\
V_0 = L_{01}\, v_0, \\
V_1 = L_{23}\, v_1,
\end{cases}
\tag{7}
$$

where $L_{03}, L_{12}, L_{01}, L_{23}$ are the edge lengths. Secondly, $\mathbf{J}^{2D}(\xi, \eta)$, the Jacobian matrix of the transformation from the physical cell to the unit cell (Fig. 3) is defined:

$$
\mathbf{J}^{2D}(\xi,\eta) = \begin{bmatrix} \frac{\partial x}{\partial \xi} & \frac{\partial x}{\partial \eta} \\ \frac{\partial y}{\partial \xi} & \frac{\partial y}{\partial \eta} \end{bmatrix} = \begin{bmatrix} \sum_n \frac{\partial \phi_n^{2D}}{\partial \xi} X_n & \sum_n \frac{\partial \phi_n^{2D}}{\partial \eta} X_n \\ \sum_n \frac{\partial \phi_n^{2D}}{\partial \xi} Y_n & \sum_n \frac{\partial \phi_n^{2D}}{\partial \eta} Y_n \end{bmatrix}.
\tag{8}
$$

The determinant of the Jacobian matrix, that will be called Jacobian is computed: $J^{2D}(\xi,\eta) = \det\left(\mathbf{J}^{2D}\right)$. The Jacobian defines the ratio between an elementary surface in the physical cell and the corresponding surface in the unit cell. The relative velocity in the unit cell is defined as (Step 3):

$$
\begin{cases}
\dfrac{\partial \xi}{\partial t} = \dfrac{(1-\xi)U_0 + \xi U_1}{J^{2D}(\xi,\eta)}, \\
\dfrac{\partial \eta}{\partial t} = \dfrac{(1-\eta)V_0 + \eta V_1}{J^{2D}(\xi,\eta)}.
\end{cases}
\tag{9}
$$

Finally (Step 4), the velocity is obtained by transforming the relative velocity (Eq. 9) to the physical coordinate system:

$$
\begin{cases}
u = \dfrac{\partial x}{\partial t} = \sum_n \left( \dfrac{\partial \phi_n^{2D}}{\partial \xi} \dfrac{\partial \xi}{\partial t} + \dfrac{\partial \phi_n^{2D}}{\partial \eta} \dfrac{\partial \eta}{\partial t} \right) X_n = \dfrac{\partial x}{\partial \xi} \dfrac{\partial \xi}{\partial t} + \dfrac{\partial x}{\partial \eta} \dfrac{\partial \eta}{\partial t}, \\
v = \dfrac{\partial y}{\partial t} = \sum_n \left( \dfrac{\partial \phi_n^{2D}}{\partial \xi} \dfrac{\partial \xi}{\partial t} + \dfrac{\partial \phi_n^{2D}}{\partial \eta} \dfrac{\partial \eta}{\partial t} \right) Y_n = \dfrac{\partial y}{\partial \xi} \dfrac{\partial \xi}{\partial t} + \dfrac{\partial y}{\partial \eta} \dfrac{\partial \eta}{\partial t}.
\end{cases}
\tag{10}
$$

**3D field**

The three-dimensional interpolation on C-grids is different for $z$- and $s$- levels.

For $z$-levels, the horizontal and vertical directions are independent. The horizontal velocities in cell $(k, j, i)$ are thus interpolated as in the 2D case, using the data at level $k$: $(u_{k,j+1,i}, u_{k,j+1,i+1}, v_{k,j,i+1}, v_{k,j+1,i+1})$, and the vertical velocity is interpolated as:

$$
w = \zeta w_0 + (1-\zeta)w_1,
\tag{11}
$$

with $w_0 = w_{k,j+1,i+1}$, and $w_1 = w_{k+1,j+1,i+1}$. The 3D indexing is again consistent with NEMO notation.

For $s$-levels, the three velocities must be interpolated at once. The three-dimensional interpolation is similar to its two-dimensional version, but it is not the straightforward extension, which would linearly interpolate the fluxes as in Eq. 9 and divide this result by the Jacobian. Indeed, in 2D the interpolation scheme is built specifically such that a uniform velocity field is exactly interpolated by Eq. 10 (see demonstration in Section 3.1). This is made possible since when developing the right hand side of Eq. 10, one obtains a numerator which is a bi-linear function of $\xi$ and $\eta$, and a denominator which is the Jacobian, precisely bi-linear in $\xi$ and $\eta$. Doing a similar approach in 3D would lead to a tri-linear numerator, but the Jacobian $J^{3D}$ is a tri-quadratic function of the coordinates $\xi$, $\eta$ and $\zeta$:

$$J^{3D}(\xi,\eta,\zeta) = \det\left(\mathbf{J}^{3D}\right), \quad \text{with } \mathbf{J}^{3D}(\xi,\eta,\zeta) = \begin{bmatrix} \frac{\partial x}{\partial \xi} & \frac{\partial x}{\partial \eta} & \frac{\partial x}{\partial \zeta} \\ \frac{\partial y}{\partial \xi} & \frac{\partial y}{\partial \eta} & \frac{\partial y}{\partial \zeta} \\ \frac{\partial z}{\partial \xi} & \frac{\partial z}{\partial \eta} & \frac{\partial z}{\partial \zeta} \end{bmatrix} = \begin{bmatrix} \sum_n \frac{\partial \phi_n^{3D}}{\partial \xi} X_n & \sum_n \frac{\partial \phi_n^{3D}}{\partial \eta} X_n & \sum_n \frac{\partial \phi_n^{3D}}{\partial \zeta} X_n \\ \sum_n \frac{\partial \phi_n^{3D}}{\partial \xi} Y_n & \sum_n \frac{\partial \phi_n^{3D}}{\partial \eta} Y_n & \sum_n \frac{\partial \phi_n^{3D}}{\partial \zeta} Y_n \\ \sum_n \frac{\partial \phi_n^{3D}}{\partial \xi} Z_n & \sum_n \frac{\partial \phi_n^{3D}}{\partial \eta} Z_n & \sum_n \frac{\partial \phi_n^{3D}}{\partial \zeta} Z_n \end{bmatrix}, \quad (12)$$

The interpolation order must then be increased as well to a quadratic function. Here, as in 2D, the velocities $u$, $v$ and $w$ are still derived from the coordinate transformation (Step 4):

$$\begin{cases} u = \frac{\partial x}{\partial t} = \sum_n \left( \frac{\partial \phi_n^{3D}}{\partial \xi} \frac{\partial \xi}{\partial t} + \frac{\partial \phi_n^{3D}}{\partial \eta} \frac{\partial \eta}{\partial t} + \frac{\partial \phi_n^{3D}}{\partial \zeta} \frac{\partial \zeta}{\partial t} \right) X_n = \frac{\partial x}{\partial \xi} \frac{\partial \xi}{\partial t} + \frac{\partial x}{\partial \eta} \frac{\partial \eta}{\partial t} + \frac{\partial x}{\partial \zeta} \frac{\partial \zeta}{\partial t}, \\ v = \frac{\partial y}{\partial t} = \sum_n \left( \frac{\partial \phi_n^{3D}}{\partial \xi} \frac{\partial \xi}{\partial t} + \frac{\partial \phi_n^{3D}}{\partial \eta} \frac{\partial \eta}{\partial t} + \frac{\partial \phi_n^{3D}}{\partial \zeta} \frac{\partial \zeta}{\partial t} \right) Y_n = \frac{\partial y}{\partial \xi} \frac{\partial \xi}{\partial t} + \frac{\partial y}{\partial \eta} \frac{\partial \eta}{\partial t} + \frac{\partial y}{\partial \zeta} \frac{\partial \zeta}{\partial t}, \\ w = \frac{\partial z}{\partial t} = \sum_n \left( \frac{\partial \phi_n^{3D}}{\partial \xi} \frac{\partial \xi}{\partial t} + \frac{\partial \phi_k^{3D}}{\partial \eta} \frac{\partial \eta}{\partial t} + \frac{\partial \phi_n^{3D}}{\partial \zeta} \frac{\partial \zeta}{\partial t} \right) Z_n = \frac{\partial z}{\partial \xi} \frac{\partial \xi}{\partial t} + \frac{\partial z}{\partial \eta} \frac{\partial \eta}{\partial t} + \frac{\partial z}{\partial \zeta} \frac{\partial \zeta}{\partial t}, \end{cases} \quad (13)$$

but this time the relative velocities interpolate the fluxes using quadratic Lagrangian functions (Step 3):

$$\begin{cases} \frac{\partial \xi}{\partial t} = \frac{\left(2\xi^2 - 3\xi + 1\right) U_0 + \left(-4\xi^2 + 4\xi\right) U_{1/2} + \left(2\xi^2 - \xi\right) U_1}{J^{3D}(\xi,\eta,\zeta)}, \\ \frac{\partial \eta}{\partial t} = \frac{\left(2\eta^2 - 3\eta + 1\right) V_0 + \left(-4\eta^2 + 4\eta\right) V_{1/2} + \left(2\eta^2 - \eta\right) V_1}{J^{3D}(\xi,\eta,\zeta)}, \\ \frac{\partial \zeta}{\partial t} = \frac{\left(2\zeta^2 - 3\zeta + 1\right) W_0 + \left(-4\zeta^2 + 4\zeta\right) W_{1/2} + \left(2\zeta^2 - \zeta\right) W_1}{J^{3D}(\xi,\eta,\zeta)}. \end{cases} \quad (14)$$

Step 2 is more complex than in the 2D case. The fluxes interpolated in Eq. 14 are the product of the velocities with the face Jacobian $J_i^{2D,f}$, as illustrated in Fig. 4. The Jacobian $J_i^{2D,f}$ is defined as follow (Weisstein, 2018):

$$J_i^{2D,f}(\xi,\eta,\zeta) = \sqrt{\left(M_{0,i}(\mathbf{J}^{3D})\right)^2 + \left(M_{1,i}(\mathbf{J}^{3D})\right)^2 + \left(M_{2,i}(\mathbf{J}^{3D})\right)^2}, \quad (15)$$

with $M_{j,i}(\mathbf{J}^{3D})$, the $j,i$ minor of $\mathbf{J}^{3D}$, i.e. the determinant of the Jacobian matrix $\mathbf{J}^{3D}$ from which row $j$ and column $i$ were removed, leading to a 2x2 matrix. Fluxes through a face that is normal to $\xi$, $\eta$ or $\zeta$ in the unit cell are computed with Jacobian $J_0^{2D,f}$, $J_1^{2D,f}$ or $J_2^{2D,f}$, respectively. Using the node indices defined in Fig. 4b, the Jacobians and their respective fluxes read

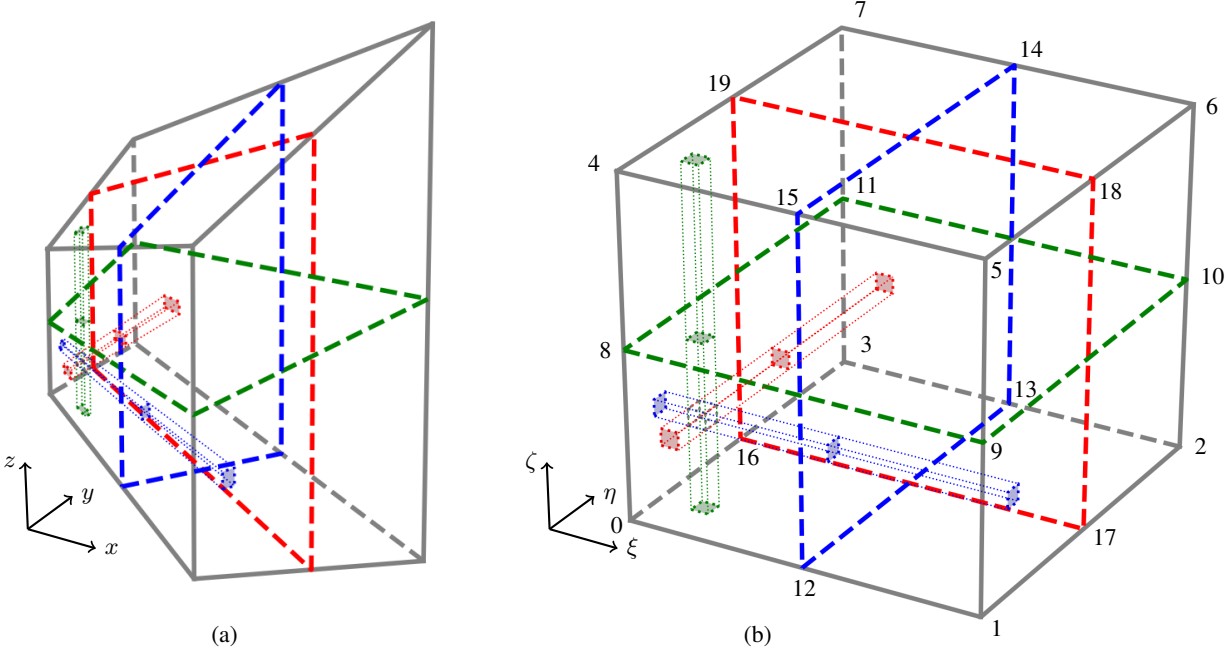

**Figure 4.** Fluxes used for 3D interpolation on a C-grid for (a) the physical cell and (b) the unit cell. The node indices in panel b are used to define the faces in Eq. 16.

then:

| face | Jacobian | flux |
|------|----------|------|
| $[0,3,7,4]$ | $J_0^{2D,f}(0,\eta,\zeta)$ | $U_0 = u_0\, J_0^{2D,f}(0,\eta,\zeta)$ |
| $[1,2,6,5]$ | $J_0^{2D,f}(1,\eta,\zeta)$ | $U_1 = u_1\, J_0^{2D,f}(1,\eta,\zeta)$ |
| $[12,13,14,15]$ | $J_0^{2D,f}(1/2,\eta,\zeta)$ | $U_{1/2} = u_{1/2}\, J_0^{2D,f}(1/2,\eta,\zeta)$ |
| $[0,1,5,4]$ | $J_1^{2D,f}(\xi,0,\zeta)$ | $V_0 = v_0\, J_1^{2D,f}(\xi,0,\zeta)$ |
| $[3,2,6,7]$ | $J_1^{2D,f}(\xi,1,\zeta)$ | $V_1 = v_1\, J_1^{2D,f}(\xi,1,\zeta)$ |
| $[16,17,18,19]$ | $J_1^{2D,f}(\xi,1/2,\zeta)$ | $V_{1/2} = v_{1/2}\, J_1^{2D,f}(\xi,1/2,\zeta)$ |
| $[0,1,2,3]$ | $J_2^{2D,f}(\xi,\eta,0)$ | $W_0 = w_0\, J_2^{2D,f}(\xi,\eta,0)$ |
| $[4,5,6,7]$ | $J_2^{2D,f}(\xi,\eta,1)$ | $W_1 = w_1\, J_2^{2D,f}(\xi,\eta,1)$ |
| $[8,9,10,11]$ | $J_2^{2D,f}(\xi,\eta,1/2)$ | $W_{1/2} = w_{1/2}\, J_2^{2D,f}(\xi,\eta,1/2)$ |

$$(16)$$

It is important to note another difference between the 2D and the 3D approaches here. While in 2D, the fluxes were simply the product of the velocity and the edge length and were independent from $\xi$ and $\eta$, this is not the case in 3D anymore. The Jacobian is a function of the relative coordinates. This is because an interface will not be evenly distorted from the physical to the unit cell. In fact, the 2D is a particular case of the 3D, where the edge length corresponds to the edge Jacobian, which is independent of the relative coordinates.

Finally, the velocities $u_0$, $u_1$, $v_0$, $v_1$, $w_0$ and $w_1$ are provided by the grid, but this is not the case for $u_{1/2}$, $v_{1/2}$, $w_{1/2}$. For ocean applications under the Boussinesq approximation, mass conservation is reduced to volume conservation (Cushman-Roisin and Beckers, 2011), leading to the continuity equation. Using this equation, the flux through faces $[12, 13, 14, 15]$ (in blue on Fig. 4), $[16, 17, 18, 19]$ (in red) and $[8, 9, 10, 11]$ (in green) can be computed if the mesh is fixed. Otherwise, for example with vertically adaptive grids, the change in volume of the mesh must be considered, which is currently not implemented in Parcels. Ignoring this term in vertically adaptive grids would lead to errors on the order of 1% (Kjellsson and Zanna, 2017).

We write here the development for face $[12, 13, 14, 15]$. The two other faces follow the same process. Let consider the hexahedron $[0, 12, 13, 3, 4, 15, 14, 7]$, the Jacobians referring to it will be noted $J^*$. The flux going through $[12, 13, 14, 15]$ reads:

$$U_{1/2}^+ = u_0\, J_0^{*2D,f}(0, 1/2, 1/2) + v_0\, J_1^{*2D,f}(1/2, 0, 1/2) - v_1\, J_1^{*2D,f}(1/2, 1, 1/2)$$
$$+ w_0\, J_2^{*2D,f}(1/2, 1/2, 0) - w_1\, J_2^{*2D,f}(1/2, 1/2, 1),$$

from which flux $U_{1/2}$ can be computed:

$$U_{1/2} = U_{1/2}^+ \frac{J_0^{*2D,f}(1, \eta, \zeta)}{J_0^{*2D,f}(1, 1/2, 1/2)}. \tag{17}$$

$U_{1/2}^+$ corresponds then to the flux through the physical face $[12, 13, 14, 15]$ and $U_{1/2}$ is the flux that should be interpolated, which results from $U_{1/2}^+$ transformed to $u_{1/2}$, the velocity at the physical face, before computing the flux in the relative cell interface. It is noteworthy that $J_0^{*2D,f}(1, \eta, \zeta) = J_0^{2D,f}(1/2, \eta, \zeta)$.

For compressible flows, fluxes $U_{1/2}^+$, $V_{1/2}^+$, $W_{1/2}^+$ cannot be computed using only half of the hexahedron. But since density is constant throughout the entire element, the actual flux $U_{1/2}^+$ can be computed as the average between the flux going through face $[12, 13, 14, 15]$ using hexahedron $[0, 12, 13, 3, 4, 15, 14, 7]$ and the one using hexahedron $[12, 1, 2, 13, 15, 5, 6, 14]$.

### 2.1.3 B-grid

The B-grid is a combination between the A- and the C- grids. It is used by OGCMs such as MOM (Griffies et al., 2004) or POP (Smith et al., 2010).

For two-dimensional fields, the velocity nodes are located on the cell vertices as in an A-grid and the tracer nodes are at the centre of the cells as in a C-grid. The velocity field is thus interpolated exactly as for an A-grid (Eq. 1), and the tracer is like in a C-grid, constant over the cell.

For a three-dimensional cell, the tracer node is still at the centre of the cell and the field is constant; the four horizontal velocity nodes are located at the middle of the four vertical edges; the two vertical velocity nodes are located at the centre of the two horizontal faces (Wubs et al., 2006). The horizontal component of the interpolated velocity in the cell thus only varies

as a function of $\xi$ and $\eta$ and the vertical component is a function of only $\zeta$:

$$
\begin{cases}
u(x,y,z) = \sum_{n=0}^{3} \phi_n^{2D}(\xi,\eta)\, u_n, \\
v(x,y,z) = \sum_{n=0}^{3} \phi_n^{2D}(\xi,\eta)\, v_n, \\
w(x,y,z) = (1-\zeta)w_0 + \zeta w_1, \\
t(x,y,z) = t_0,
\end{cases}
\tag{18}
$$

with the local indices ($u_0$, $u_1$, $u_2$, $u_3$) corresponding to global indices ($u_{k,j,i}$, $u_{k,j,i+1}$, $u_{k,j+1,i+1}$, $u_{k,j+1,i}$) and similarly for the $v$ field. $w_0$ and $w_1$ correspond to $w_{k,j+1,i+1}$ and $w_{k+1,j+1,i+1}$ and tracer $t_0$ to $t_{k,j+1,i+1}$. In Parcels v2.0, this field interpolation is only available for $z$-levels.

## 2.2 Implementation into Parcels

### 2.2.1 Fields

**General Structure**

Parcels relies on a set of `Field` objects, combined in a `FieldSet`, to interpolate different quantities at the particle location. As explained in Section 2.1, a field is discretised on a grid. In Parcels v2.0, four grid objects are defined: `RectilinearZGrid`, `RectilinearSGrid`, `CurvilinearZGrid` and `CurvilinearSGrid`.

The main variables of the grids are the time, depth, latitude and longitude coordinates. Longitude and latitude are defined as vectors for rectilinear grids and 2D arrays for curvilinear ones. The depth variable is defined as a vector for $z$-level grids. A two-dimensional horizontal grid is simply a `RectilinearZGrid` or a `CurvilinearZGrid` in which the depth variable is empty. For $s$-levels, the depth is defined as a 3D array or a 4D array if the grid moves vertically in time. The time variable can be empty for steady state fields. Note that models such as Hycom using hybrid grids combining $z$- and $\sigma$-levels are treated by Parcels as general `RectilinearSGrid` or `CurvilinearSGrid` objects, for which the depth of the $s$-levels is provided.

A `Field` has an `interp_method` attribute, which is set to `linear` for fields discretised on a A-grid, `bgrid_velocity`, `bgrid_w_velocity` and `bgrid_tracer` for horizontal and vertical velocity and tracer fields on a B-grid and `cgrid_velocity` and `cgrid_tracer` for velocity and tracer on a C-grid. Note that a `nearest` interpolation method is also available, which should not be used for B- and C- grids.

Parcels can load field data from various input formats. The most common approach consists in reading netCDF files using the `FieldSet.from_netcdf()` method or one of its derivatives. However, Python objects such as xarray or numpy arrays can also be loaded using `FieldSet.from_xarray_dataset()` or `FieldSet.from_data()`, respectively.

Loading a long time series of data often requires a significant memory allocation, which is not always available on the computer. The previous Parcels version circumvented the problem by loading the data step by step. Using `deferred_load` flag which is set by default, this process is fully automated in v2.0 and allows to use long time series while under the hood the time steps of the data are loaded only when they are strictly necessary for computation.

**Meta-field objects**

In Parcels, a variety of other objects enable to easily read a field. In this section, we describe the new objects recently added to the framework.

The first object is the `VectorField`, that jointly interpolates the two or three components of a vector field such as velocity.
This object is not only convenient but necessary, since $u$ and $v$ fields are both required to interpolate the zonal and meridional velocity in the C-grid curvilinear discretisation.

Another useful object is the `SummedField`, which is a list of fields that can be summed to create a new field. The fields of the `SummedField` do not necessarily share the same grid. For example, this object can be used to create a velocity which is the sum of surface water current and Stokes drift. This object has no other purpose than simplifying greatly the kernels defining the particle dynamics.

The fields do not necessarily have to cover the entire region of interest. If a field is interpolated outside its boundary, an `ErrorOutOfBounds` is raised, which leads to particle deletion except if this error is processed through an appropriate kernel. A sequence of various fields covering different regions, which may overlap or not, can be interpolated under the hood by Parcels with a `NestedField`. In this case, the fields composing the `NestedField` must represent the same physical quantity. The `NestedField` fields are ranked to set the priority order in which they must be interpolated.

Available data are not always provided with the expected units. The most frequent example is the velocity given in m s$^{-1}$ while the particle position is in degrees. The same problem occurs with diffusivities in m$^2$ s$^{-1}$. `UnitConverter` objects allow to convert automatically the units of the data. The two examples mentioned above are defined into Parcels and other `UnitConverter` objects can be implemented by the user for other transformations.

### 2.2.2 Kernels

The kernels define the particle dynamics (Lange and van Sebille, 2017). Various built-in kernels are already available in Parcels. `AdvectionRK4`, `AdvectionRK45` and `AdvectionEE` implement the Runge-Kutta 4, Runge-Kutta-Fehlberg and explicit Euler integration schemes for advection. While other explicit discrete time schemes can be defined, analytical integration schemes (Blanke and Raynaud, 1997; Chu and Fan, 2014) are not yet available in Parcels.

`BrownianMotion2D` and `SpatiallyVaryingBrownianMotion2D` implement different types of Brownian motion available as kernels. Custom kernels can be defined by the user for an application-dependent dynamics.

The kernels are implemented in Python but are executed in C for efficiency (Lange and van Sebille, 2017) even if a full Python mode is also available. However, the automated translation of the kernels from Python to C somehow limits the freedom in the syntax of the kernels. For advanced kernels, the possibility to call a user-defined C library is available in version 2.0.

## 3 Validation

### 3.1 Uniform velocity on a 2D C-grid

In this section, we prove that the C-grid interpolation preserves exactly a uniform velocity in a quadrilateral. To do so, let us define a uniform velocity $\mathbf{u} = (u, v)$ and a quadrilateral with $x$ coordinates $[X_0, X_1, X_2, X_3]$ and $y$ coordinates $[Y_0, Y_1, Y_2, Y_3]$. On such an element, the velocities $u_0$, $u_1$, $v_0$, $v_1$ are the scalar product of $\mathbf{u}$ and $\mathbf{n}$, the unit vector normal to the edge, and the fluxes $U_0$, $U_1$, $V_0$, $V_1$ are the velocities multiplied by the edge lengths, leading to:

$$U_0 = u\,(Y_3 - Y_0) - v\,(X_3 - X_0)$$
$$U_1 = u\,(Y_2 - Y_1) - v\,(X_2 - X_1)$$
$$V_0 = u\,(Y_0 - Y_1) - v\,(X_0 - X_1)$$
$$V_1 = u\,(Y_3 - Y_2) - v\,(X_3 - X_2).$$

Therefore, developing Eq. 9 results in:

$$J^{2D}\frac{\partial \xi}{\partial t} = \quad u\frac{\partial y}{\partial \eta} - v\frac{\partial x}{\partial \eta}$$
$$J^{2D}\frac{\partial \eta}{\partial t} = -u\frac{\partial y}{\partial \xi} + v\frac{\partial x}{\partial \xi},$$

and then applying Eq. 10:

$$\frac{\partial x}{\partial t} = u$$
$$\frac{\partial x}{\partial t} = v,$$

independently from $\xi$ and $\eta$. For the 3D case, the same result is obtained numerically. It can be evaluated using the simple Python C-grid interpolator code available at https://doi.org/10.5281/zenodo.3253697.

### 3.2 $z$- and $s$- level C-grid compatibility

As mentioned above, the horizontal and vertical directions in grids using $z$-levels are completely decoupled, such that horizontal velocity can be computed as for a 2D field, and vertical interpolation is computed linearly. But a $z$-level grid is a particular case of an $s$-level grid. We show that the 3D C-grid interpolator reduces to the simpler $z$-level C-grid when $Z_0 = Z_1 = Z_2 = Z_3$ and $Z_4 = Z_5 = Z_6 = Z_7$.

First, for $z$-levels, it is noteworthy that:

$$\sum_{i=0}^{7} \frac{\partial \phi_i^{3D}}{\partial \xi} X_i = \sum_{i=0}^{3} \frac{\partial \phi_i^{2D}}{\partial \xi} X_i = \frac{\partial x}{\partial \xi},$$

and similarly for $\frac{\partial x}{\partial \eta}, \frac{\partial y}{\partial \xi}, \frac{\partial y}{\partial \eta}$. $\mathbf{J}^{3D}$ is:

$$\mathbf{J}^{3D} = \begin{bmatrix} \frac{\partial x}{\partial \xi} & \frac{\partial x}{\partial \eta} & 0 \\ \frac{\partial y}{\partial \xi} & \frac{\partial y}{\partial \eta} & 0 \\ 0 & 0 & Z_4 - Z_0 \end{bmatrix},$$

and $J^{3D} = J^{2D}(Z_4 - Z_0)$. All the fluxes through the vertical faces reduce to the product of the velocity, the horizontal edge length and the element height, as for $U_0$:

$U_0 = u_0 J_0^{2D,f} = u_0 (Z_4 - Z_0) \sqrt{\left(\frac{\partial x}{\partial \eta}\right)^2 + \left(\frac{\partial y}{\partial \eta}\right)^2} = u_0 (Z_4 - Z_0) \sqrt{(X_3 - X_0)^2 + (Y_3 - Y_0)^2}.$

The fluxes through the horizontal faces are:

$$W_0 = w_0 J^{2D}, \quad W_1 = w_1 J^{2D}.$$

Therefore, the inner fluxes results in:

$$U_{1/2} = 0.5 (U_0 + U_1), \tag{19}$$

and similarly for $V_{1/2}$ and $W_{1/2}$. The first two lines of Eq. 13 reduce to Eq. 10 and the third line is:

$$\frac{\partial z}{\partial t} = (Z_4 - Z_0) \frac{\partial \zeta}{\partial t}. \tag{20}$$

Finally using Eq. 19, Eq. 14 becomes:

$$\begin{cases} \frac{\partial \xi}{\partial t} = \frac{(1 - \xi) U_0 + \xi U_1}{(Z_4 - Z_0) J^{2D}}, \\ \frac{\partial \eta}{\partial t} = \frac{(1 - \eta) V_0 + \eta V_1}{(Z_4 - Z_0) J^{2D}}, \\ \frac{\partial \zeta}{\partial t} = \frac{(1 - \zeta) W_0 + \zeta W_1}{(Z_4 - Z_0) J^{2D}}, \end{cases}$$

from which first two lines correspond to Eq. 9 and third combined with Eq. 20 reduces to Eq. 11.

## 15   4   Simulating the sensitivity of North West European continental shelf floating microplastic distribution

Microplastic (MP) is transported through all marine environments and has been observed in large quantities both at coastlines (e.g. Browne et al., 2011) and open seas (e.g. Barnes et al., 2009), at the surface and the sea bed. It represents potential risks to the marine ecosystem that cannot be ignored (Law, 2017). At a global scale, high concentrations are reported in the subtropics (Law et al., 2010) but also in the Arctic (Obbard et al., 2014). Various studies have already modelled the accumu-
lation of MP in the Arctic (van Sebille et al., 2012, 2015; Cózar et al., 2017), highlighting the MP transport from the North

Atlantic and the North Sea. Meanwhile, at smaller scales, other studies have focused focused on marine litter in the Southern part of the North Sea (Neumann et al., 2014; Gutow et al., 2018; Stanev et al., 2019) and have included diffusion and wind drift to their model as well as used a higher resolution.

Here we study how the modelled accumulation of floating MP in the Arctic depends on the incorporation of physical processes and model resolutions used for the Southern part of the North Sea. Parcels is used to evaluate the sensitivity of the floating MP distribution under those constrains. To do so, virtual floating MP particles are released off the Rhine and Thames estuaries and tracked for three years. The floating MP distribution is then compared with the trajectories of passive 3D particles, which are not restricted to stay at the sea surface. Note that this section is not meant as a comprehensive study of the MP transport off the North Sea, but rather an application of the new features implemented into Parcels, in both two and three dimensions.

## 4.1 Input data

We study the influence of the different physical processes impacting surface currents like density- and wind- driven currents, tidal residual currents and Stokes drift, but also the impact of mesh resolution and diffusion. The data come from various data sets (Fig. 5), described in this section. Links to access the data are provided in the code and data availability section.

### 4.1.1 NEMO

The main data we use are ORCA0083-N006 and ORCA025-N006, which are standard sets-up from NEMO (Madec et al., 2016), an ocean circulation model forced by reanalysis and observed flux data: the Drakkar Forcing Set (Dussin et al., 2016). The forcings consist of wind, heat and freshwater fluxes at the surface.

The data are available globally at resolutions of $1/4°$ (ORCA025-N006) and $1/12°$ (ORCA0083-N006). They are discretised on an ORCA grid (Madec and Imbard, 1996), a global ocean tripolar grid, which is curvilinear. The mesh is composed of 75 $z$-levels and the variables are positioned following a C-grid. The temporal resolution is 5 days.

### 4.1.2 North West shelf reanalysis

The North West shelf reanalysis (Mahdon et al., 2017) is an ocean circulation flow data set based on the Forecasting Ocean Assimilation Model 7 km Atlantic Margin Model, which is a coupling of NEMO for the ocean with the European Regional Seas Ecosystem Model (Blackford et al., 2004). The reanalysis contains tidal residual currents.

The data are freely available on the Copernicus Marine Environment Monitoring Service (CMEMS). They have a resolution of about 7 km ($1/9°$ lon x $1/15°$ lat), from ($40°$ N, $20°$ W) to ($65°$ N, $13°$ E), with a temporal resolution of 1 day. The data are originally computed on a C-grid, but are re-interpolated on the tracer nodes to form an A-grid data set with $z$-levels, which is available on CMEMS.

The data, which will be referred to as NWS, do not cover the entire modelling region, such that a `NestedField` is used to interpolate it within the available region (green zone in Fig. 5), and use the NEMO data for particles outside that region.

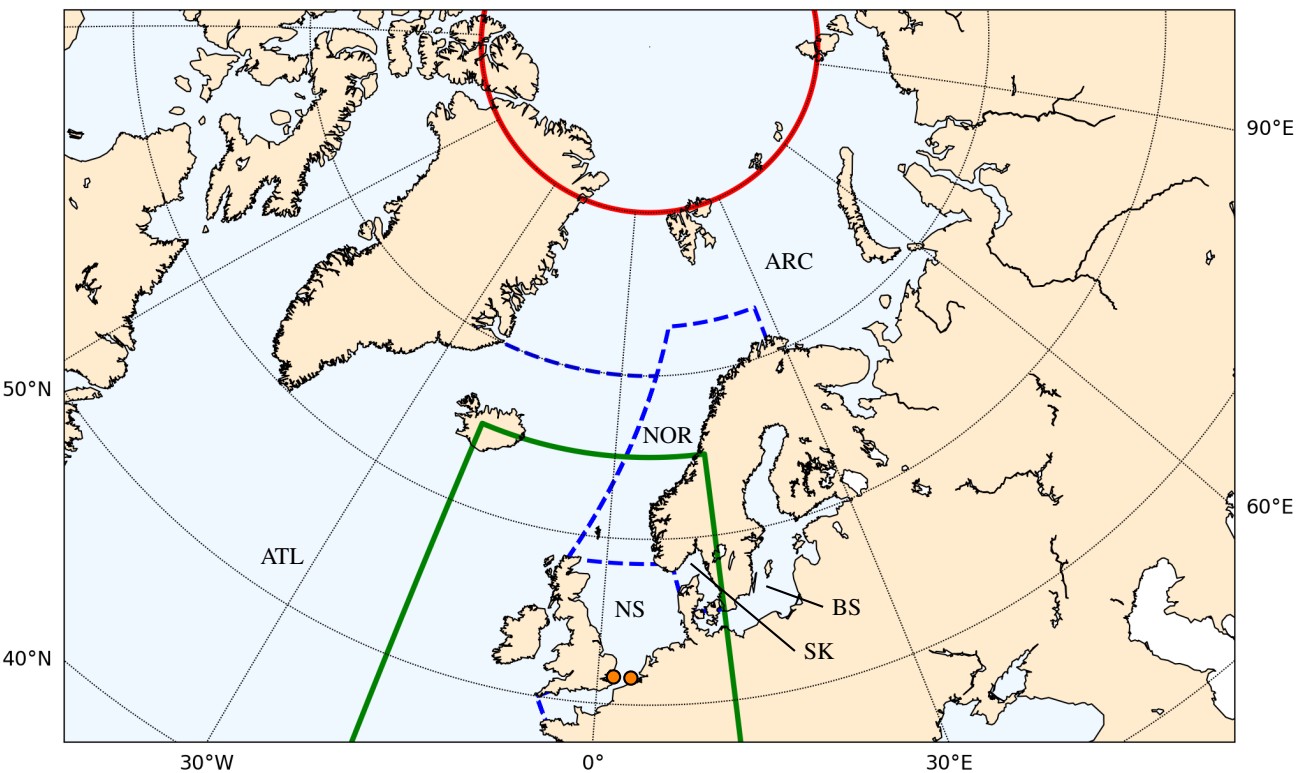

**Figure 5.** Spatial coverage of the OGCM data used to study North Sea microplastic transport. NEMO data are available globally. NWS data are available for the North Sea region (green boundaries) and WaveWatch III data are available South of 80° N (red boundary). The particle releasing locations are the Rhine and Thames estuaries (orange dots) and the region is split into six zones: North Sea (NS), Skagerrak and Kattegat (SK), Baltic Sea (BS), Norwegian Coast (NOR), Arctic Ocean (ARC) and Atlantic Ocean (ATL), that are separated by the dashed blue lines.

### 4.1.3 WaveWatch III

Stokes drift, i.e. the surface residual current due to waves, was obtained from WaveWatch III (Tolman et al., 2009), that was run using wind forcings from the NCEP Climate Forecast System Reanalysis (CFSR, Saha et al., 2010). The data have a spatial resolution of 1/2°, extending until 80° N, and a temporal resolution of 3 hours.

5 **4.2 Simulations**

Six simulations are run in the following configurations: (a) NEMO hydrodynamics at a 1/12° resolution, (b) NEMO at a 1/4° resolution, (c) NWS hydrodynamics in the North Sea nested into NEMO 1/12°, (d) NEMO 1/12° coupled with WaveWatch III Stokes drift, (e) NEMO 1/12° with diffusion and finally (f) NEMO hydrodynamics at a 1/12° resolution in which the particles are not constrained to the sea surface.

Every day of year 2000, 100 particles are released in the mouth of the Thames estuary and 100 more particles in mouth of the Rhine, before being tracked for three years. For the two 2D runs, (a) to (e), the particles are released at the sea surface and follow the horizontal surface currents. For the 3D run (f), the particles are released at 0.1, 0.5 and 1 m depths and follow the 3D NEMO flow field.

The diffusion, which parametrises the unresolved processes, is modelled as a stochastic zeroth-order Markov model (van Sebille et al., 2018). The diffusion parameter is proportional to the mesh size, exactly as in the study of North Sea marine litter by Neumann et al. (2014):

$$D = D_0 \, (l/l_0)^{4/3} \,, \tag{21}$$

with $D_0 = 1 \text{ m}^2 \text{ s}^{-1}$ the reference diffusivity, $l$ the square root of the mesh size and $l_0 = 1$ km. This formulation leads to a same
order of magnitude diffusivity as the constant value of Gutow et al. (2018).

    The beaching of MP is non-negligible in the North Sea (Gutow et al., 2018) even if it is still poorly understood and often ignored (Neumann et al., 2014) or resulting from the low-resolution current and wind conditions (Gutow et al., 2018). Here we distinguish two flow types, the first based on NEMO and NWS data which has impermeable boundary conditions at the coast, and the second which includes Stokes drift and diffusion, thus allowing beaching.

For numerical reasons, due to the integration time step of 15 minutes and the Runge-Kutta 4 scheme, it is theoretically possible that particles beach even with NEMO or NWS data. This could happen for example in a region of coastal downwelling, since the particles are forced to stay at the surface and could be constantly transported towards the coast. The particle dynamics is thus implemented using separate kernels. At each time step, the particle position is first updated following NEMO or NWS advection. Then the particle is checked to still be located in a wet cell, otherwise it is pushed back to the sea using an artificial
current. In a second step, the Stokes or diffusion kernels are run, where if the particle beaches, it stops moving. In a final step, the particle age is updated. The kernel code as well as all the scripts running and post-processing the simulations are available at https://doi.org/10.5281/zenodo.3253693.

    To compare the simulations, the Parcels raw results, consisting of particle position, age and beaching status exported every two days, are post-processed into the following maps and budgets.

The particle density (Fig. 6) is computed as the number of particles per square kilometre, averaged over the third year of particle age. Note that the absolute value of the concentration is not particularly meaningful, since it is simply proportional to the number of particles released.

    To analyse the particle path, the ocean is discretised into cells of 1/4° longitude x 1/8° latitude resolution, and for each cell the fraction of particles that has visited at least once the cell is computed (Fig. 7).

To study the temporal dynamics of the particles, the region is divided into six zones (Fig. 5): North Sea, Skagerrak and Kattegat, Baltic Sea, Norwegian Coast, Arctic Ocean and Atlantic Ocean, and the evolution of the distribution of the particles in those zones is computed (Fig. 8). The time axis represent the particle age in years.

    Finally, the integrated vertical distribution (Fig. 9) of the particles as a function of the latitude is computed for the 3D run. For this profile, the domain is divided in bins of 0.5° latitude by 5 m depth and every two days, each particle is mapped to

the cell it belongs to, leading to the integrated vertical distribution. Note the linear scale for the upper 50 m depth, and the logarithmic scale for the full vertical profile.

Animations showing the particle dynamics are available in the article supplementary materials.

## 4.3 Results

The results show various minor and major differences between the scenarios.

While NEMO 1/12° and NEMO 1/4° show similar dynamics for the first year (Fig. 8), the Norwegian fjords have a higher trapping role in the 1/4° resolution run, even if the plastic does not beach in both runs. This increased particle trapping could be a consequence of the data lower resolution that results in a reduced horizontal velocity shear, while a strong shear layer behaves as a barrier isolating the coast from the open waters (Delandmeter et al., 2017). This is also observed in the

10 supplementary materials animations, in which the main particle path is observed further from the coast for the NEMO 1/12° run. As a consequence of the trapping, the amount or MP reaching the Arctic is reduced in NEMO 1/4°. This run also produces significantly lower densities North of 80° N. Although none of the runs do resolve coastal dynamics, because of the low temporal and space resolution and the lack of tides, they show important differences. Since no validation was achieved for those MP simulations, there is no reason to argue that the 1/12° resolution is high enough to simulate MP dynamics, but this

resolution is similar to other studies of plastic litter in the region (Gutow et al., 2018).

The main differences of using NWS result from the dynamics during the first year, when the particles are located South of 65° N, explaining the lack of differences in the Arctic region in Fig. 6a and Fig. 6c and Fig. 7a and Fig. 7c. The residence time in the North Sea is increased relatively to NEMO 1/12°, and different peak events occur in the Skagerrak and Kattegat, resulting in final concentrations of 68% in the Arctic and 27% in the Norwegian coast, respectively 7% higher and 8% lower

than with NEMO 1/12° (Fig. 8).

Including Stokes drift has a major impact on MP dynamics in the North West European continental shelf, due to prevailing westerly winds (Gutow et al., 2018), with close to 90% of the plastic staying in the North Sea, 9% beaching in the Norwegian coast and less than 0.25% reaching the Arctic. Particles are beaching very quickly, with 90% in less than 3.5 months, and 99% within 10 months. While those numbers are not validated here, we can still point out that even if Stokes drift has an important

contribution on surface dynamics for large scales (Onink et al., 2019), using it on smaller scales needs a proper validation. Especially the boundary condition should be treated with care. It has a large impact in this application where the particles are released next to the coast.

The parametrisation of sub-grid scales and diffusion is still an important field of research in the Lagrangian community, but it is generally agreed that it cannot be neglected. In this application, we observe how adding diffusion impacts the fate of MP.

The amount of MP reaching the Arctic is reduced by 68% compared to NEMO 1/12°, with large accumulation in the North Sea and Norwegian coast, but not in the Skagerrak and Kattegat. Overall, the proportion of beached particles increases linearly to 73% during the first year, before slowly reaching 83% during the next two years.

Maintaining the MP at the surface is a strong assumption: biofouling, degradation and hydrodynamics affect the plastic depth, which impacts its lateral displacement. In the 3D particle run (Fig. 6f), we do not take all the processes driving MP

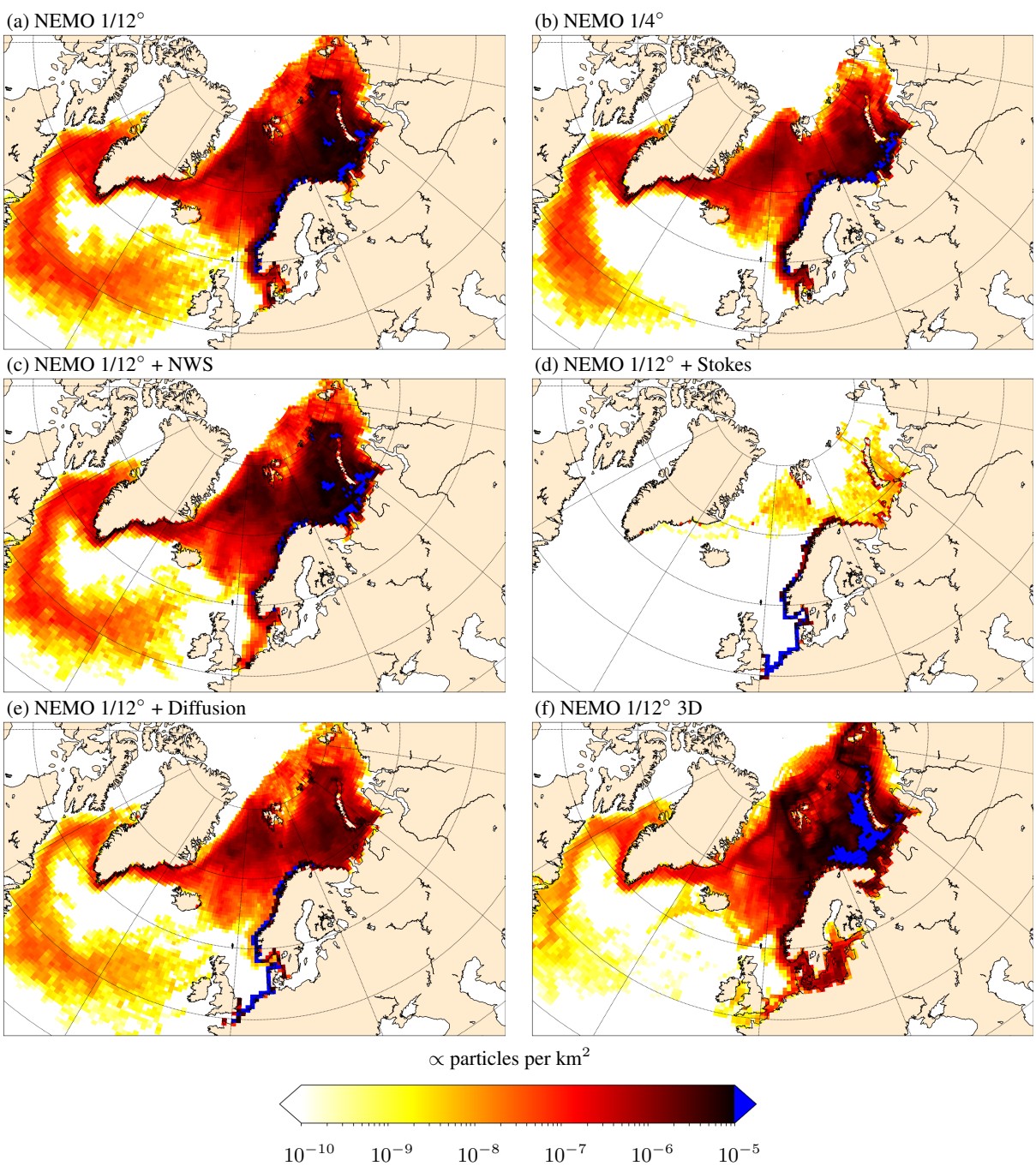

**Figure 6.** Density of floating microplastic (a-e) and passive 3D particles (f), averaged over the third year of particle age, for the different simulation scenarios: (a) NEMO 1/12°, (b) NEMO 1/4°, (c) NWS nested into NEMO 1/12°, (d) NEMO 1/12° coupled with Stokes drift from WaveWatch III, (e) NEMO 1/12° coupled with diffusion, (f) NEMO 1/12° 3D. Note the logarithmic scale.

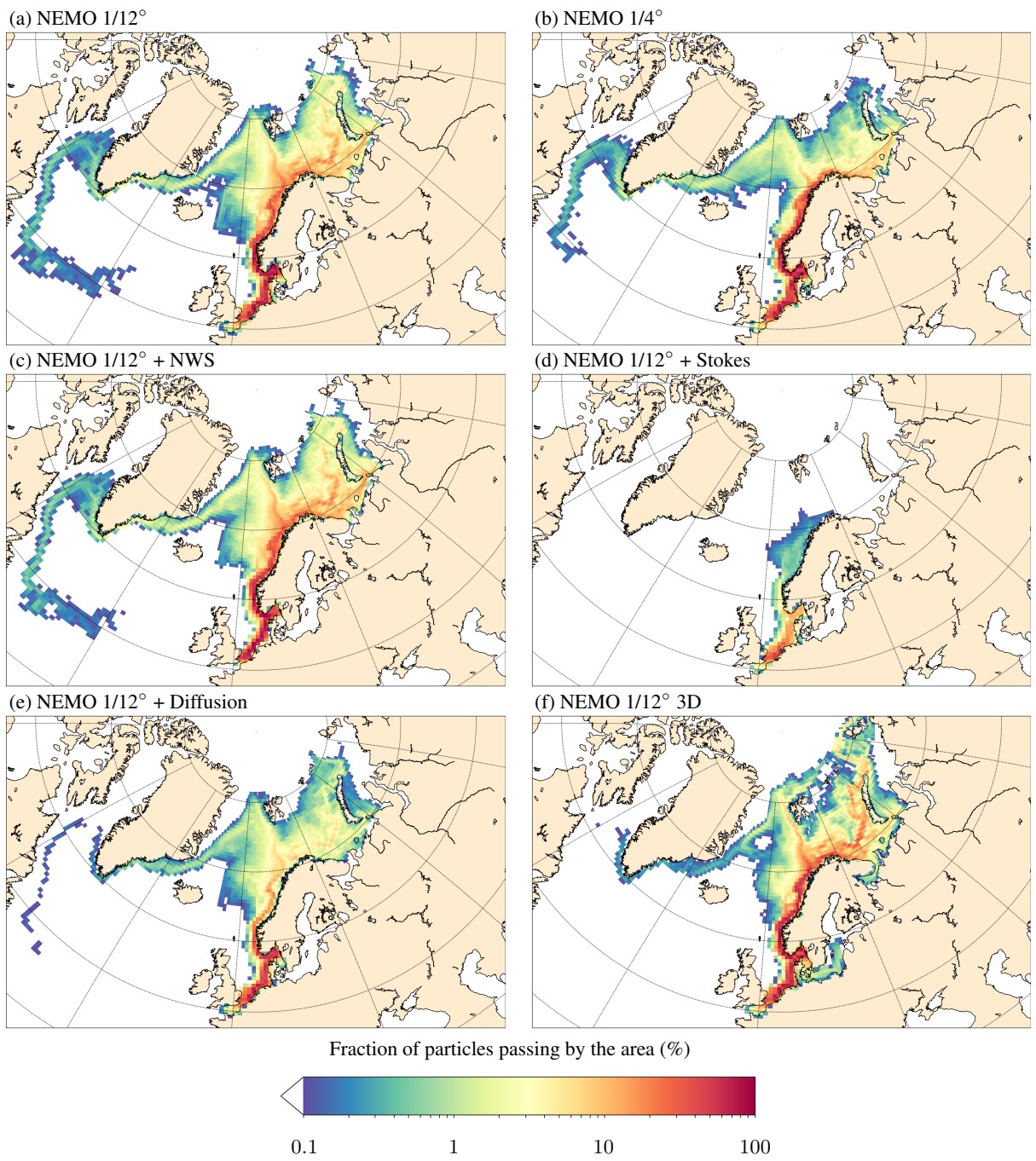

**Figure 7.** Fraction of floating microplastic (a-e) and passive 3D particles (f) originally released in the Thames and Rhine estuaries reaching the domain. It is computed for each cell of a $1/4°$ longitude x $1/8°$ latitude grid, as the proportion of the particles that have visited at least once the cell. Note the logarithmic scale.

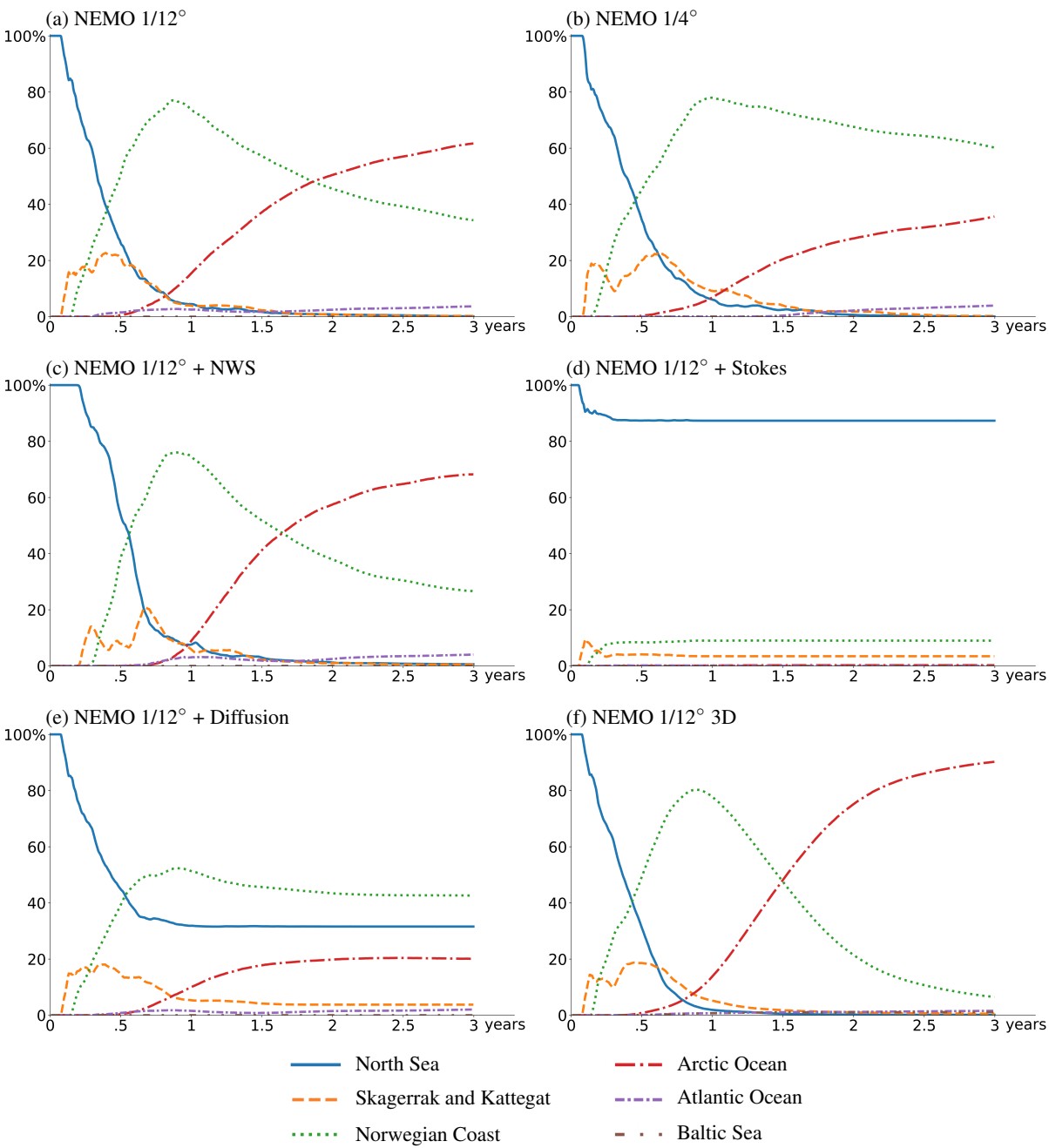

**Figure 8.** Evolution of the distribution of floating microplastic (a-e) and passive 3D particles (f) in the six zones (Fig. 5) as a function of particle age.

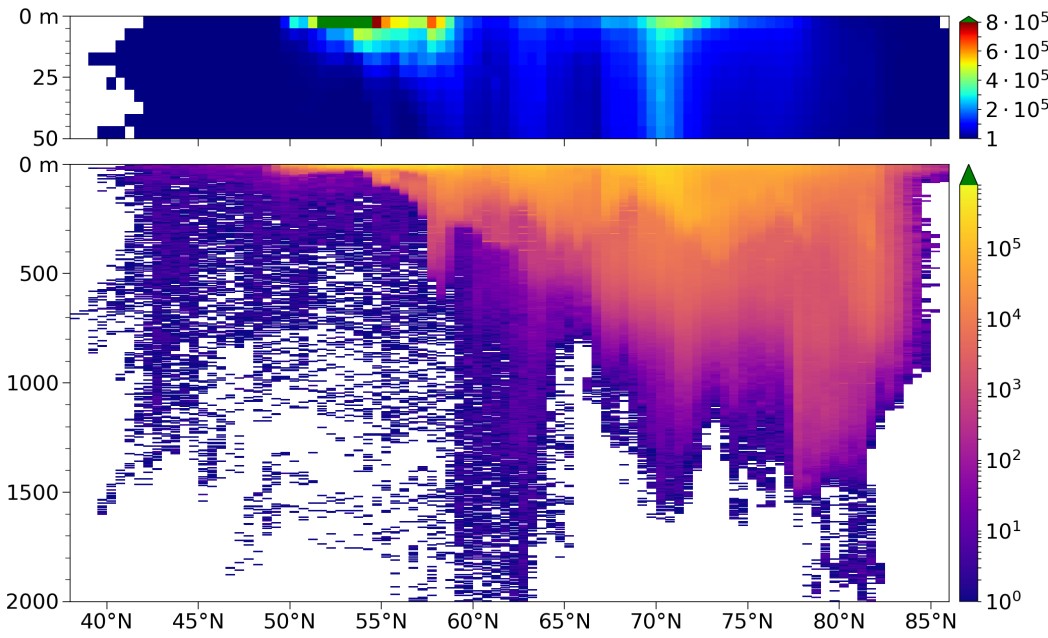

**Figure 9.** Particle integrated vertical distribution for the 3D NEMO 1/12° simulation. Note the linear scale for the upper 50 m depth, and the logarithmic scale for the full profile.

vertical dynamics, whose parametrisations are currently being developed by the community, but simply model the path of passive particles following the three-dimensional current. Passive particles do not accumulate along the coast as the floating MP (Fig. 6f). This results in an increased transport towards the Arctic (Figs. 7f and 8f). There are overall higher concentrations in the Arctic, including North of 80°N and in the Eastern part of the Kara Sea. Around 1% of the passive particles end up in the

5 Baltic Sea, crossing the Skagerrak and Kattegat using favourable deeper currents (Gustafsson, 1997), while such transport was negligible for floating MP plastic. The vertical distribution of the particles is also analysed (Fig. 9): the majority of the particle, originally released in shallow waters, is still observed close to the surface: 21%, 63% and 97% of the particle records are above 10 m, 100 m and 500 m depths, respectively. The concentration vertical gradient decreases while the latitude increases, with almost no gradient in the Polar region. Passive particles have a significantly different path then surface ones, highlighting the

10 importance of understanding better the vertical dynamics of the plastic to improve the accuracy of its distribution modelling.

This brief study of the sensitivity of North Sea floating MP distribution is an illustration of how Parcels is used to gather and compare flow fields from a multitude of data sets in both two and three dimensions, which was made possible by the development of the different field interpolation schemes and meta-field objects. To validate the MP dynamics observed, it is essential to couple such numerical study with an extensive field study.

# 5 Conclusions

Parcels, a Lagrangian ocean analysis framework, was considerably improved since version 0.9, allowing to read data from multiple fields discretised on different grids and grid types. In particular, a new interpolation scheme for curvilinear C-grids was developed and implemented into Parcels v2.0. This article described this new interpolation as well as the other schemes available in Parcels, including A-, B- and C- staggered grids, rectilinear and curvilinear horizontal meshes and $z$- and $s$- vertical levels. Numerous features were implemented, including meta field objects, which were described here.

Parcels v2.0 was used to simulate the dynamics of the North West European continental shelf floating microplastic, virtually released during one year off the Thames and Rhine estuaries, before drifting towards the Arctic, and the sensitivity of this transport to various physical processes and numerical choices such as mesh resolution and diffusion parametrisation. While those simulations do not provide a comprehensive study of microplastic dynamics in the area, they highlight key points to consider and illustrate the interest of using Parcels for such modelling.

The next step in Parcels development will involve increasing the model efficiency and developing a fully parallel version of the Lagrangian framework.

*Code and data availability.*

– Parcels code: The code for Parcels is licensed under the MIT licence and is available through GitHub at https://www.github.com/OceanParcels/parcels. The version 2.0 described here is archived at Zenodo at https://doi.org/10.5281/zenodo.3257432. More information is available on the project webpage at http://www.oceanparcels.org.

– Interpolation code: Independently from Parcels, a simple Python code is also implementing all the C-grid interpolation schemes developed in this paper. It is available at https://doi.org/10.5281/zenodo.3253697.

– North Sea floating MP simulations: All the scripts running and post-processing the North Sea MP simulations are available at https://doi.org/10.5281/zenodo.3253693.

– NEMO data: The NEMO N006 data are kindly provided by Andrew Coward at NOC Southampton, UK, and can be downloaded at http://opendap4gws.jasmin.ac.uk/thredds/nemo/root/catalog.html.

– North West shelf reanalysis data are provided by the Copernicus Marine Environment Monitoring Service (CMEMS). They can be downloaded at http://marine.copernicus.eu/services-portfolio/access-to-products/?option=com_csw&view=details&product_id=NORTHWESTSHELF_REANALYSIS_PHY_004_009.

– WaveWatch III data come from the Ifremer Institute, France. They can be downloaded at ftp://ftp.ifremer.fr/ifremer/ww3/HINDCAST/GLOBAL/.

*Author contributions.* Philippe Delandmeter and Erik van Sebille developed the code and wrote the paper jointly.

*Competing interests.* The authors declare that they have no conflict of interest.

*Acknowledgements.* Philippe Delandmeter and Erik van Sebille are supported through funding from the European Research Council (ERC) under the European Union Horizon 2020 research and innovation programme (grant agreements nos. 715386 and 821926). The North Sea microplastic simulations were carried out on the Dutch national e-infrastructure with the support of SURF Cooperative (project no. 16371).
5   This study has been conducted using E.U. Copernicus Marine Service Information. We thank Henk Dijkstra for the fruitful discussions and Andrew Coward for providing the ORCA0083-N006 and ORCA025-N006 simulation data.

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
