# Peer review of "The Parcels v2.0 Lagrangian framework: new field interpolation schemes"

_Geoscientific Model Development, 2018_

## Referee Comment (RC1) · Knut-Frode Dagestad (Referee) · 17 Feb 2019

General comments

————————-

This manuscript describes the upgrades leading to version 2.0 of the Parcels lagrangian drift code, with emphasis on new field interpolation schemes. A short section is devoted to mathematical validation of the new schemes, and a longer section is devoted to a demonstration study of floating microplastics. The manuscript is well written, with few typos. Figures, in particular the field diagrams, are very clear and useful. In

general the paper demonstrates and documents nicely the accuracy, power and flexibility of the Parcels software, and is as such worthy publication In GMD. My only major concern about this paper, as commented below, is that the study of floating microplastics does not address very directly the new field interpolation schemes, which are the major focus of this paper.

Specific comments

————————-

- The new interpolation schemes addresses z and sigma as vertical coordinates. However, the widely used ocean model HYCOM uses a hybrid combination, with z-coordinates near the surface and sigma-coordinates below. Please comment whether the existing schemes are applicable to native HYCOM-output, or if such an extension would be simple/feasible, and eventually whether it is planned for the (near) future.

- It could also be commented whether there are any specific plans to support unstructured grids in the (near) future.

- The paper addresses various types of fields, but does not mention file formats. E.g. would any CF-compatible netCDF-file be directly ingestible by Parcels 2.0? What would be the approach for using model-specific output formats, including non-netCDF?

- In section 2.1.1 velocities are referred to as "zonal" and "meridional", and in Section 2.2.1 it is said that longitude and latitude are 1D arrays for rectilinear grids. Does this imply that only lon-lat (Plate Caree) and Mercator grids are supported, and not other projections such as e.g. polar stereographic?

- Is the coastline always considered to be land-pixels of the ocean model, or does Parcels 2.0 also support using e.g. vector coastlines such as GSHHS? This affects e.g. the question about impermeability (i.e. that ocean currents do not have an onshore component). The interpolation schemes assures impermeability at the coast, which is a good property, but this would be difficult to assure if the same ocean models is not

used for the landmask. And with nested grids, each ocean model would then need to use their own coastline to assure impermeability?

- The WaveWatch Stokes drift is only available up to 80 degree North, nevertheless the simulations including Stokes drift shows particles further north. Is the Stokes drift simply neglected northwards of 80 degrees, instead of deactivating the particles?

- With the Nemo 1/4 degree model, more MP is trapped along the Norwegian Coast than with the 1/12 model. Can you explain why this is the case?

- The sensitivity study of Section 4 addresses "floating microplastics". It is not stated specifically, but I assume this means that the MP is considered to be at the very surface all the time? If so, this has some implications which should be commented. E.g. have in situ measurements (e.g. Kukulka, 2012, Kooi, 2016) shown that MP particles are mixed deeper in the water column with increasing wind speeds. This implies that the real Stokes drift is less than the surface Stokes drift, which is presumably used in this study?

- Figure 8 shows e.g. that the effect of adding Stokes drift, is that more particles are kept at the Norwegian coast, and less is advected into the Arctic Ocean. However, it is very hard to see this from Figures 6 a) and d), probably due to the logarithmic scale. What is the reason for choosing a logarithmic scale on Figure 6 and 7?

- Section 4.2 explains that impermeability condition applies to the ocean current advection, but that Stokes drift and diffusion allow stranding. However, if diffusion is regarded as unresolved (sub-grid) ocean currents, these should in principle also not have any onshore component. This would imply that the amount of stranding is overestimated.

- It would also be nice to have some short comments about the implementation of the new interpolation schemes. Are these programmed in Python, or in C? Are they programmed at a lowest level (i.e. the equations as shown in this paper), or are some external higher level Python libraries used? Any comments about computational time/performance would also be welcome, either in general terms (fast, very fast, slow. . .) or as numerical metrics.

- As mentioned under the General Comments, the study of floating microplastics does not appear to test specifically the new interpolation schemes, apart from demonstrating that fields of different grids may be used and combined. NEMO input is on a curvilinear C-grid, whereas CMEMS input is on an A-grid (rectilinear?), but the vertical coordinates (z or sigma) are not mentioned. However, if the plastics is considered to be at the surface all the time, the vertical interpolation is not even used in this study. A more direct validation/test, would e.g. be to compare two simulations with 3D-drift with the CMEMS/NEW data set in respectively native coordinates (C-grid curvilinear), and the regridded (A-grid rectilinear) data as used in this study. In addition to quantifying the differences (hopefully small?) of the spatial distribution/drift for such a case, it would be interesting the get an idea about the difference in computational time, where using pre-regridded data would be expected to be faster.

Technical comments

——————————

- Figure1: it could be commented that all 4 combinations of horizontal and vertical grids are possible.

- Both in Section 2.1.1 and 2.1.2 there are unnumbered sub-headings names "2D field" and "3D field". This could lead to confusion when jumping back and forth between pages/sections.

- Section 2.1.1, line 15. The meaning of this sentence is unclear: "The interpolation must use local information in the cells."

- Section 2.2.1 says that data is read lazily, which is a nice property. Is this based on external libraries such as dask, or is it explicitly programmed in Parcels?

- There are links to the interpolation code, which is said to be independent of Parcels.

Does this mean that it is implemented as a stand-alone library which is used by Parcels, or is it (also) directly included in the Parcels codebase?

- Throughout the paper, Microplastics is abbreviated as MP, which is fine. However, for the figure captions it might be useful to be explicit, as figures are sometimes used/read out of direct context of the paper.

- Figure 7: Could also comment here that scale is logarithmic.

- Page 14, line 14: please give the Ifremer FTP address (or refer to the data availability section at the end)

- Page 15, line 9: "consisting at" -> "consisting of".

- Page 16, line 4: could specify that 1/4 degree is longitude, and 1/8 degree is latitude.

- Page 16m line 12: "even if this" -> "even if the"

- The North West shelf reanalysis is referred to as "CMEMS". However, CMEMS provides a lot of different data, also including NEMO. Thus I would recommend using a more specific reference, such as e.g. "NWS".

—————————————————————

---

## Referee Comment (RC2) · Joakim Kjellsson (Referee) · 20 Feb 2019

**Summary**
The paper describes the new version of Parcels, v2.0. The new version includes new interpolation schemes for tracing particles which allow for various vertical coordinates and staggered grids. As the paper presents these new and necessary features of Parcels I recommend it should be published, but only after some re-working of the text and also using a 3D test case rather than a 2D case.

**Major comments**

[Figure]

The authors spend quite some time deriving and explaining the new interpolation schemes for z and s coordinates on staggered grids, which is a new feature of Parcels. I'm therefore a bit puzzled that the showcase in the Results section is only for surface drift. I would strongly recommend the authors to change the showcase to some example with subsurface 3D flow, so that we can see the z or s coordinates in action.

The writing is in need of revising (see specific comments further down). In particular, I struggled with Page 6, Line 23 to Page 9 Line 20, which did not flow well and was at times confusing. This section needs a bit of re-writing and re-structuring. I also found Section 2.1.1 to be very abstract and I had to wait until Section 3 before the methods to be described in a more practical sense. I would strongly urge the authors to either put Section 3 directly after 2.1.1 or somehow merge the two sections so that the definitions of interpolation schemes are directly followed by how it is done in practice.

**Minor overall comments**

On the large scale I find that the authors over-use the word "different" as a synonym for "various", and I often found that the word could simply be omitted to make the paper easier to read.

Parcels is referred to as being developed to meet the exa-scale challenge, when velocity fields and tracer fields become massive and traditional Lagrangian codes will struggle. However, none of the examples in the paper are what I would refer to as "very large data sets", and there are no results regarding Parcels run time, memory use etc. I would therefore change the focus a little bit and re-phrase the introduction and also throughout the paper to describe Parcels as flexible and user-friendly, which

seems to be the big advantage of using Parcels, rather than focusing too much on computational efficiency.

**Specific comments**

Line 19: I would re-write to say "can, in turn, be used to analyse the global ocean dynamics given the flow field from the model." Followed by "The flow field can also be taken from observations, e.g. land-based measurement . . . "

Page 2:
Line 2: "and many other types", "etc."
Line 26: "We then validate . . . "
Line 29: "the results."

Page 5:
I'm wondering if Fig 2 is really necessary. The staggering of grids is also shown in Fig 3, and indices could also be added to Fig 3, thus making Fig 2 redundant.

Page 6:
Line 15: "in the cells, and interpolating"
Line 16: "formulation. For instance, such interpolation"
Line 18: While I enjoy citations, it it enough to just cite Jonsson et al. 2015 (the Tracmass code) and Doos et al 2017 (a thorough model description paper).
Line 23: If I understand this section correctly, you calculate fluxes on the cell faces, then interpolate fluxes to particle position, and then interpolate cell face area to the particle position, and divide flux by area to find velocity? The section starts by defining

the velocity and ends by defining the Jacobian, which makes it a bit confusing for the reader to follow how velocities are interpolated from the model grid to the particle position. It would make more sense to start by defining the fluxes U,V, then how they are interpolated to the particle position, and then describe how velocities are found.
Line 24: "(Fig 2b). Velocities are not found by linear interpolation but, like in finite-volume schemes, they are approximated by linearly interpolating the fluxes (U0,U1,V0,V1) at the cell faces (fig 3b) and dividing by the cell face area."
Line 26: Should it be ". . . the velocity and any position (x,y) is derived as a function . . . "?

Line 5: These are the velocities on the model grid? The section should end with an expression for how u,v are found. Line 9-11: The last two sentences seem out of place. Instead, you could add ", where indices are chosen to conform with the NEMO model (Madec et al)" on Page 6, Line 24.
Line 17: $l$ is the model vertical index? Conforming with NEMO model?

Page 8:
Line 1: what is meant by "do not resolve exactly a uniform velocity"? Do you mean "do not result in a uniform velocity"
Line 6: remove "different"

Page 7:
Line 5: "and their respective fluxes are"
Line 6: I like this Table. Could you do the same thing for the 2D case and also add to Fig 3? It would be a lot shorter, but I think it could be informative.
Line 9-12: Replace with "We can compute the fluxes through grid faces [12,13,14,15] (in blue, Fig 4), [16,17,18,19] (in red), and [8,9,10,11] (in green) using the continuity equation. The flux through [12,13,14,15] is . . . "

Line 13-14: Is this only for fixes z coordinates? In the case of $z^*$ or $\sigma$ coordinates, the cell thickness varies in time which must be taken into account. If the time-varying part is taken into account here, please explain how.

Line 16: What does the "+" superscript mean? What is the difference between $U^+$ and $U$?

Page 10:
Line 5: "four grid objects"
Line 13: "which should not be used for C-grids"
Line 16: ". . . describe the new objects which were added . . . "
Line 27: ". . . regions, which may overlap or not . . . "
Line 28: remove "different"
Line 29: ". . . order in which they . . . "

Page 11:
Line 3: ". . . is the velocity given in . . . "

Line 5: ". . . transported through . . . "
Line 10-13: ". . . studies have focused on marine littler in the southern part of the North Sea (Neumann) and have included diffusion and wind drift to their model as well as used a higher resolution."
Line 23: "NEMO-N006" is not a standard I am familiar with, at least it does not ship with NEMO v3.6. Are you referring to the ORCA0083-N006 simulation, which is similar to ORCA0083-N001 used by Grist et al 2014 and Kjellsson & Zanna 2017?
Line 26: ". . . at horizontal resolutions of nominally 1/4 and 1/12."
Line 26: What is the vertical coordinate used, i.e. which interpolation scheme is used here?

Line 7: Full disclosure: this is the reason why I'm often sceptical about CMEMS data for particle modelling. All data is interpolated from the model grid to some other grid and not necessarily in a conservative way. Could you say a few words here about how this interpolation was done by CMEMS. Would you get identical results if the CMEMS data came on the native C-grid from the NEMO model?

Line 9: "...data, which will be ..."

Line 10: Again, which vertical coordinate is used? Is this also interpolated by CMEMS?

Line 28-29: After "(Gutow 2018).": "Here we distinguish two flow types, the first based on NEMO and CMEMS data which has impermeable boundary conditions at the coast, and the second which includes Stokes drift and diffusion thus allowing beaching."

Line 5: "...advection." Line 6: "...are run, where if the particle beaches, it stops moving."

Line 5: what is meant by "travelled at least once by a cell"? That the cell has been visited by at least one particle?

Line 10: remove "different"

Line 15: "...no validation of mesoscale dispersion has been done for those simulations there ...". I am fairly sure Andrew C has done some validation (AMOC strength, AABW volume etc.), but probably not for particle dispersion near the grid scale.

Line 20-22: I don't fully understand this sentence. By "differences generated in the first year" you mean "within the first year we see more transport into the Kattegat and Skaggerak leaving fewer particles for transport along the Norwegian coast"? There are some differences along the Norwegian coast and Barents Sea (less deep blue

regions for Fig 6c).
Line 27-28: "...how adding diffusion impacts the face of MP."
Line 28: "reduced by 68%" is in relation to NEMO 1/12?

I don't understand the caption of the figure. Do you mean "fraction of particles visiting each different region at least once" or "for each grid cell, fraction of particles that have visited that grid cell".?

––––––––––––––––––––––––––––––

---

## Author Comment (AC1) · 4 Apr 2019

We would like to thank Dr Knut-Frode Dagestad for his careful reading and its constructive comments. Please find our replies below.

Philippe Delandmeter and Erik van Sebille

***General comments***

*This manuscript describes the upgrades leading to version 2.0 of the Parcels lagrangian drift code, with emphasis on new field interpolation schemes. A short section is devoted to mathematical validation of the new schemes, and a longer section is devoted to a demonstration study of floating microplastics. The manuscript is well written, with few typos. Figures, in particular the field diagrams, are very clear and useful. In general the paper demonstrates and documents nicely the accuracy, power and flexibility of the Parcels software, and is as such worthy publication In GMD. My only major concern about this paper, as commented below, is that the study of floating microplastics does not address very directly the new field interpolation schemes, which are the major focus of this paper.*

Thank you. In the answer below, we address the different comments, with a special care about the main one concerning the relevance of the North Sea microplastic application.

**Specific comments**

*The new interpolation schemes addresses z and sigma as vertical coordinates. However, the widely used ocean model HYCOM uses a hybrid combination, with z-coordinates near the surface and sigma-coordinates below. Please comment whether the existing schemes are applicable to native HYCOM-output, or if such an extension would be simple/feasible, and eventually whether it is planned for the (near) future.*

State-of-the-art ocean models use a large variety of vertical coordinate systems and vertical mesh discretisations. In Parcels, we differentiate them into two categories: $z$- and $s$- levels. The $z$-levels are defined as a vector of the depth of the mesh levels. The $s$-levels are a 3D-array of the mesh level depths, depending on lon-lat position and vertical levels. The combination of $z$- and $\sigma$- levels in HYCOM is then naturally read by Parcels, that considers them as general $s$-levels. We added this discussion in the

revised manuscript (page 11, line 16 of the revised manuscript).

*It could also be commented whether there are any specific plans to support unstructured grids in the (near) future.*

Incorporating unstructured grids involves three main developments: (1) interpolating a field in a cell; (2) finding in which cell the particle is located; (3) a new data structure. We already do (1) for general quadrilaterals (2D) and hexahedra (3D), but other common meshes (triangles and prisms) would be required to support unstructured grids. To find where is our particle (2), we use some properties of the mesh, that do not hold on unstructured meshes. The current data structure is built for structured data and should then be generalised (3). The Parcels team is still a relatively small group and there is no plan to support unstructured grids in the near future.

*The paper addresses various types of fields, but does not mention file formats. E.g. would any CF-compatible netCDF-file be directly ingestible by Parcels 2.0? What would be the approach for using model-specific output formats, including non-netCDF?*

The most common way to provide input data to Parcels is indeed to use netCDF files, although other formats such as numpy or xarray objects are also available. The Parcels input data loading functions have been improved since version 0.9 (Lange et al., 2017), but their general structure has remained the same. Following your comment, we have added a note on the input data format (page 11, line 22).

*In section 2.1.1 velocities are referred to as "zonal" and "meridional", and in Section 2.2.1 it is said that longitude and latitude are 1D arrays for rectilinear grids. Does this imply that only lon-lat (Plate Caree) and Mercator grids are supported, and not other projections such as e.g. polar stereographic?*
Parcels is not specifically designed for lon-lat coordinates, even if those are the coordinates in most of the applications. As long as the coordinates (which are abusively called lon, lat, depth in Parcels) are in the same system as the velocities, there is nothing to do. If it is not the case, velocities can be rescaled using UnitConverters as it is explained in page 12 line 16. UnitConverters between lon/lat and metric systems are already implemented, but other can be added by the user. Finally, if the velocity need a more specific transformation (for example a rotation), this needs to be done within a kernel, that can also be added by the user.

*Is the coastline always considered to be land-pixels of the ocean model, or does Parcels 2.0 also support using e.g. vector coastlines such as GSHHS? This affects e.g. the question about impermeability (i.e. that ocean currents do not have an on-shore component). The interpolation schemes assures impermeability at the coast, which is a good property, but this would be difficult to assure if the same ocean models is not used for the landmask. And with nested grids, each ocean model would then need to use their own coastline to assure impermeability?*

This is a very interesting comment. Parcels does not support directly vector coastlines. We prefer to read the data as they were generated by the OGCM for a better consistency. But as you pointed out, when multiple fields from different datasets are used, this can result in situations where the particle is considered on the beach according to one field, but not the other. The solution to this situation is problem dependent. This is again where Parcels kernels show their importance. For example in the 2D north sea microplastic application, we do have multiple fields with different coastlines. We define a particle as on beach if it is out of NEMO or NWS boundaries. A kernel is naturally dealing this situation (https://github.com/OceanParcels/Parcelsv2. 0PaperNorthSeaScripts/blob/master/northsea_mp_kernels.py). If the user wants to use GSHHS data for example, it can easily take advantage of the Python language of Parcels, to rasterize the data on a grid and provide it to the model.

*The WaveWatch Stokes drift is only available up to 80 degree North, nevertheless the simulations including Stokes drift shows particles further north. Is the Stokes drift simply neglected northwards of 80 degrees, instead of deactivating the particles?*

Indeed, the Stokes drift was simply neglected northwards of 80°N. However, when revising the code, we spotted a bug that was highly affecting the Stokes dynamics: due to a typo, the time step had been removed from the Stokes dynamical kernel. We have now fixed this bug and revised entirely the code to ensure that such error was only present there. After this fix, the dynamics of the MP including Stokes drift have changed drastically. The new results are discussed in the revised manuscript. The particles going North are now negligible.

*With the Nemo 1/4 degree model, more MP is trapped along the Norwegian Coast than with the 1/12 model. Can you explain why this is the case?*

This is interesting. Indeed, more MP is trapped along the Norwegian coast with the low resolution model. This is not a consequence of beaching particles, since any particle accidentally beaching would be pushed back towards the water, and we observed that such beaching was negligible. When observing the particle dynamics now available in the supplementary materials, we observe that the particle main path stays further away from the coast with the HR data. This might be a consequence of the HR data, which produce higher lateral shear in the surface velocities, that acts as a barrier protecting the beach from open waters. We discuss this difference in the revised manuscript (page 18, line 6).

*The sensitivity study of Section 4 addresses "floating microplastics". It is not stated specifically, but I assume this means that the MP is considered to be at the very surface all the time? If so, this has some implications which should be commented. E.g. have*

*in situ measurements (e.g. Kukulka, 2012, Kooi, 2016) shown that MP particles are mixed deeper in the water column with increasing wind speeds. This implies that the real Stokes drift is less than the surface Stokes drift, which is presumably used in this study?*

Yes, the MP stays the whole time at the surface in those simulations. It is a strong assumption, which is taken in many studies of the MP distribution (Onink et al., 2019). Better approximation of the plastic dynamics are currently being developed, and we are working on this as well. But this paper, which focuses on the numerics, simply aims to show how the distribution of the surface MP could be affected by the use of different datasets. As we have written, it is meant as an illustration of the Parcels framework, not a comprehensive study of MP in the North Sea.

Following the comments of both reviewers, we also run 3D passive particle dynamics in this revised version. This also shows the effect of 3D dynamics on particle transport (see also new figure showing vertical distribution), even though this second simulation is not specifically dedicated towards plastic transport, which do not follow passively the 3D hydrodynamics.

*Figure 8 shows e.g. that the effect of adding Stokes drift, is that more particles are kept at the Norwegian coast, and less is advected into the Arctic Ocean. However, it is very hard to see this from Figures 6 a) and d), probably due to the logarithmic scale. What is the reason for choosing a logarithmic scale on Figure 6 and 7?*

We have first post-processed the data using linear scales (see Figs 1 and 2 of this comment). Such graphs give a lot of details in a certain range of concentration, but the multiscale dimension of the distribution is hard to see. With the logarithmic scale, we have exactly the opposite situation. That is why we chose the second option, but have Figure 8 which, as you pointed, provides information that is hardly seen on the maps.

*Section 4.2 explains that impermeability condition applies to the ocean current advection, but that Stokes drift and diffusion allow stranding. However, if diffusion is regarded as unresolved (sub-grid) ocean currents, these should in principle also not have any onshore component. This would imply that the amount of stranding is overestimated.*

Ocean current data (NEMO and NWS) have impermeable boundary condition, that we keep in our Lagrangian simulation. Stokes drift data do not have impermeable boundary condition, such that we allow Stokes drift to beach the particles. The third source of transport is the diffusion. It parameterises unresolved processes, which includes sub-grid scales and the coastal boundary layer, the last one allowing beaching. We then chose a boundary condition that allows beaching. Since the simulation has not been calibrated against observation, it should not be seen as a realistic simulation of the MP transport, especially close to the coast, but as a study of the importance of the diffusion term.

*It would also be nice to have some short comments about the implementation of the new interpolation schemes. Are these programmed in Python, or in C? Are they programmed at a lowest level (i.e. the equations as shown in this paper), or are some external higher level Python libraries used? Any comments about computational time/performance would also be welcome, either in general terms (fast, very fast, slow. . .) or as numerical metrics.*

The schemes are implemented at the lowest level, as they are described in the manuscript. Parcels can be run either in a Python-C coupled way (for efficiency) or fully in Python (development mode), such that the schemes are implemented twice: in Python and in C. Interpolation schemes for A-grid are available in the scipy library, but this is not the case with our new C-grid interpolator, such that we do not use any external interpolation library.

While this manuscript focuses on the interpolation schemes, we are currently working

on the model performance and develop a parallel implementation of the code. So far the overall CPU time is dominated by IO communication, such that we do not see any difference in terms of performance between A- and C- grids.

*As mentioned under the General Comments, the study of floating microplastics does not appear to test specifically the new interpolation schemes, apart from demonstrating that fields of different grids may be used and combined. NEMO input is on a curvilinear C-grid, whereas CMEMS input is on an A-grid (rectilinear?), but the vertical coordinates (z or sigma) are not mentioned. However, if the plastics is considered to be at the surface all the time, the vertical interpolation is not even used in this study. A more direct validation/test, would e.g. be to compare two simulations with 3D-drift with the CMEMS/NEW data set in respectively native coordinates (C-grid curvilinear), and the regridded (A-grid rectilinear) data as used in this study. In addition to quantifying the differences (hopefully small?) of the spatial distribution/drift for such a case, it would be interesting the get an idea about the difference in computational time, where using pre-regridded data would be expected to be faster.*

After reading your comments and the ones from Dr Kjellsson, we have included a new 3D simulation, that advects passive particles interpolating the NEMO $1/12°$ data, which are discretised on a C-grid with $z$-levels. This new simulation is not a proof that the interpolation scheme dynamics, the analytical proofs being provided in Section 3, but an illustration of a 3D run in Parcels.

We did not directly compare our simulation on a C-grid, with a simulation with the same regridded data: first, we did not have such data; but more importantly, as we commented to Dr Kjellsson, this would not be a proof of our interpolation schemes dynamics, but a validation of the regridding algorithm. If A-grid data are generated in a conservative form with correct boundary conditions, there will still be some difference between the two dynamics, but this difference will be small. The advantage of our approach is that no regridding is necessary, but the users can interpolate the data they

have access for the different types of grids.

***Technical comments***

*Figure1: it could be commented that all 4 combinations of horizontal and vertical grids are possible.*

Indeed. We added such comment.

*Both in Section 2.1.1 and 2.1.2 there are unnumbered sub-headings names "2D field" and "3D field". This could lead to confusion when jumping back and forth between pages/sections.*

While we agree with your comment, technical instructions from Copernicus require to not use paragraphs but only subsubsections. We have emphasized at the beginning of 2.1.1 and 2.1.2 that we consider separately the 2D and 3D cases.

*Section 2.1.1, line 15. The meaning of this sentence is unclear: "The interpolation must use local information in the cells."*

Indeed, this sentence was not clear and have been rephrased.

*Section 2.2.1 says that data is read lazily, which is a nice property. Is this based on external libraries such as dask, or is it explicitly programmed in Parcels?*

The Python-C structure of Parcels currently prevents us from using lazy loading functionalities of xarray and dask. Indeed, the data needs to be loaded into memory before being manipulated by the C-library. We still implement lazy loading, by only loading the data time steps when required. This is directly implemented into Parcels.

*There are links to the interpolation code, which is said to be independent of Parcels. Does this mean that it is implemented as a stand-alone library which is used by Parcels, or is it (also) directly included in the Parcels codebase?*

The simple interpolation code provides information to the reader who wants to see a Python implementation of the schemes developed in this paper. In Parcels, the interpolation schemes are directly implemented in the code, independently from the small library.

*Throughout the paper, Microplastics is abbreviated as MP, which is fine. However, for the figure captions it might be useful to be explicit, as figures are sometimes used/read out of direct context of the paper.*

You are right. We modified the figure captions in that sense.

*Figure 7: Could also comment here that scale is logarithmic.*

Done.

*Page 14, line 14: please give the Ifremer FTP address (or refer to the data availability section at the end)*

We added at the beginning of the data section that all links are provided in the data availability section.

*Page 15, line 9: "consisting at" $->$ "consisting of".*

Done, thank you.

*Page 16, line 4: could specify that 1/4 degree is longitude, and 1/8 degree is latitude.*

We modified the sentence.

*Page 16m line 12: "even if this" − > "even if the"*

Done.

*The North West shelf reanalysis is referred to as "CMEMS". However, CMEMS provides a lot of different data, also including NEMO. Thus I would recommend using a more specific reference, such as e.g. "NWS".*

You are right. We now refer to the North West shelf reanalysis data as NWS.

(a) NEMO 1/12°  (b) NEMO 1/4°

(c) NEMO 1/12° + NWS  (d) NEMO 1/12° + Stokes

(e) NEMO 1/12° + Diffusion  (f) NEMO 1/12° 3D

∝ particles per km²

0   $2\cdot10^{-6}$   $4\cdot10^{-6}$   $6\cdot10^{-6}$   $8\cdot10^{-6}$   $10^{-5}$

**Fig. 1.**

[Figure]

(a) NEMO 1/12°  (b) NEMO 1/4°

(c) NEMO 1/12° + NWS  (d) NEMO 1/12° + Stokes

(e) NEMO 1/12° + Diffusion  (f) NEMO 1/12° 3D

Proportion of plastic passing by the area (%)

0.1    20    40    60    80    100

**Fig. 2.**

---

## Author Comment (AC2) · 4 Apr 2019

We would like to thank Dr Joakim Kjellsson for his careful reading and its constructive comments. Please find our replies below.

Philippe Delandmeter and Erik van Sebille

*Summary*

*The paper describes the new version of Parcels, v2.0. The new version includes new interpolation schemes for tracing particles which allow for various vertical coordinates and staggered grids. As the paper presents these new and necessary features of Parcels I recommend it should be published, but only after some re-working of the text and also using a 3D test case rather than a 2D case.*

Thank you. Please find our answers to your different comments below.

**Major comments**

*The authors spend quite some time deriving and explaining the new interpolation schemes for z and s coordinates on staggered grids, which is a new feature of Parcels. I'm therefore a bit puzzled that the showcase in the Results section is only for surface drift. I would strongly recommend the authors to change the showcase to some example with subsurface 3D flow, so that we can see the z or s coordinates in action.*

Following your comments and the ones from Dr Dagestad, we haved added a simulation with 3D passive particles in the North Sea to compare surface and 3D transport in the North West European continental shelf. In this run, the particles dynamics follow the NEMO $1/12°$ data, which are discretised on a curvilinear C-grid with $z$-levels.

*The writing is in need of revising (see specific comments further down). In particular, I struggled with Page 6, Line 23 to Page 9 Line 20, which did not flow well and was at times confusing. This section needs a bit of re-writing and re-structuring. I also found Section 2.1.1 to be very abstract and I had to wait until Section 3 before the methods to be described in a more practical sense. I would strongly urge the authors to either put Section 3 directly after 2.1.1 or somehow merge the two sections so that the definitions of interpolation schemes are directly followed by how it is done in practice.*

Section 2.1 introduced a lot of definitions and notations, which made it hard to read. Based on your comments (here and below), we have restructured the section to help the reader along with the construction of the interpolation schemes, highlighting the main four steps:

1. define a mapping between the physical cell and a unit cell;

2. compute the fluxes on the unit cell interfaces, as a function of the velocities on the physical cell interfaces;

3. interpolate those fluxes to obtain the relative velocity;

4. transform the relative velocity to the physical velocity.

Before we were describing the scheme by starting with what we want (the velocity at given $x$, $y$, $z$) and progressively building the variables we need, ending with the gridded input data. We have reversed that order for the 2D case, starting from what we have and reaching what we want. This structure follows the one in Section 3, that you commented as easier to understand. For the 3D case, we keep the original order since there is no point to introduce the different fluxes ($U_0$, $U_{12}$, $U_1$, $V_0$, ...) before motivating why we need them, but we keep referring to the four main steps of the interpolation. Section 3 is a validation of the schemes, and therefore appears after the complete description.

*Minor overall comments*

*On the large scale I find that the authors over-use the word "different" as a synonym for "various", and I often found that the word could simply be omitted to make the paper easier to read.*

We have removed the unnecessary occurrences of this word as well as others for an easier reading.

*Parcels is referred to as being developed to meet the exa-scale challenge, when velocity fields and tracer fields become massive and traditional Lagrangian codes will struggle. However, none of the examples in the paper are what I would refer to as "very large data sets", and there are no results regarding Parcels run time, memory use etc. I would therefore change the focus a little bit and re-phrase the introduction and also throughout the paper to describe Parcels as flexible and user-friendly, which seems to be the big advantage of using Parcels, rather than focusing too much on computational efficiency.*

The aim of Parcels is to build an efficient and flexible framework for Lagrangian ocean analysis (Lange et al., 2017). We simply remind this in the paper introduction, then do not insist on it since it is not the point of the paper.

*Specific comments*

*Page 1:*
*Line 19: I would re-write to say "can, in turn, be used to analyse the global ocean dynamics given the flow field from the model." Followed by "The flow field can also be taken from observations, e.g. land-based measurement... "*

We have modified the sentence.

*Page 2:*
*Line 2: "and many other types", "etc." Line 26: "We then validate... "*
*Line 29: "the results."*

Done

*Page 5:*
*I'm wondering if Fig 2 is really necessary. The staggering of grids is also shown in Fig 3, and indices could also be added to Fig 3, thus making Fig 2 redundant.*

Fig. 3 is already quite busy with other information. We preferred to not overload it, and keep the global indexing separately, in Fig. 2.

*Page 6:*
*Line 15: "in the cells, and interpolating"*
*Line 16: "formulation. For instance, such interpolation"*
*Line 18: While I enjoy citations, it it enough to just cite Jonsson et al. 2015 (the Tracmass code) and Doos et al 2017 (a thorough model description paper).*

Done.

*Line 23: If I understand this section correctly, you calculate fluxes on the cell faces, then interpolate fluxes to particle position, and then interpolate cell face area to the particle position, and divide flux by area to find velocity? The section starts by defining the velocity and ends by defining the Jacobian, which makes it a bit confusing for the reader to follow how velocities are interpolated from the model grid to the particle position. It would make more sense to start by defining the fluxes U,V, then how they are*

*interpolated to the particle position, and then describe how velocities are found. Line 24: "(Fig 2b). Velocities are not found by linear interpolation but, like in finite-volume schemes, they are approximated by linearly interpolating the fluxes (U0,U1,V0,V1) at the cell faces (fig 3b) and dividing by the cell face area."*
*Line 26: Should it be ". . . the velocity and any position (x,y) is derived as a function..."?*

The interpolation schemes consist in (1) building a mapping between the physical and the unit cell; (2) computing the fluxes on the cell edges (in 2D) or faces (in 3D); (3) interpolating the relative velocity using the fluxes and the Jacobian of the transformation; (4) transform the velocity to the physical coordinates. We have rewritten the section to highlight the important parts of the interpolation process, reordering the description as you suggested.

*Page 7:*
*Line 5: These are the velocities on the model grid? The section should end with an expression for how u,v are found. Line 9-11: The last two sentences seem out of place. Instead, you could add ", where indices are chosen to conform with the NEMO model (Madec et al)" on Page 6, Line 24. Line 17: l is the model vertical index? Conforming with NEMO model?*

Those were the relative velocities. After the restructuring of the section, we now end with the expression for $u$ and $v$ as you suggested. The comment on the NEMO indexing was also moved following your comment. Yes, we use NEMO conforming indexing in both horizontal and vertical directions. $l$ was the vertical index, since $k$ was used somewhere else, but we have changed it such that we use the more common $i$, $j$ and $k$ notations for the grid indexing.

*Page 8:*

*Line 1: what is meant by "do not resolve exactly a uniform velocity"? Do you mean "do not result in a uniform velocity"*
*Line 6: remove "different"*

We meant that if the C-grid field was representing a uniform velocity, it was not possible to obtain for all $x$, $y$ and $z$ that velocity while interpolating the field with a linear interpolation scheme in 3D. We have reformulated that sentence.

*Page 9:*
*Line 5: "and their respective fluxes are"*

Done

*Line 6: I like this Table. Could you do the same thing for the 2D case and also add to Fig 3? It would be a lot shorter, but I think it could be informative.*

For the 2D case, the situation is much easier since there are only 4 fluxes, and the Jacobians reduce to the simple edge length. We have added the 2D computed fluxes definition in new Eq. 7.

*Line 9-12: Replace with "We can compute the fluxes through grid faces [12,13,14,15] (in blue, Fig 4), [16,17,18,19] (in red), and [8,9,10,11] (in green) using the continuity equation. The flux through [12,13,14,15] is..."*

We have reformulated that sentence.

*Line 13-14: Is this only for fixes z coordinates? In the case of z\* or $\sigma$ coordinates, the cell thickness varies in time which must be taken into account. If the time-varying part is taken into account here, please explain how.*

Thank you for pointing that comment. Indeed, this is only valid for a fixed mesh. For a moving mesh, the mesh expansion should be added to the continuity equation, which we don't do in Parcels. We have added this note to the document, with also a reference to Kjellsson and Zanna 2017, that quantifies the error induced if a moving mesh is assumed as fixed.

*Line 16: What does the "+" superscript mean? What is the difference between U+ and U?*

There is an important difference between $U_{12}$ and $U_{12}^+$. $U_{12}^+$ corresponds to the flux going through the physical interface. From this flux, the velocity $u_{12}$ is computed as the flux $U_{12}^+$ divided by the interface area, which corresponds to the Jacobian evaluated at $\eta = 0.5$, $\zeta = 0.5$. Then as for the other interfaces, the flux to be interpolated $U_{12}$ is the product of $u_{12}$ with the Jacobian $J_0^{2D,f}(0.5, \eta, \zeta)$. This distinction results from the fact that the Jacobian varies as a function of the relative coordinates. Volume is not distorted evenly within the cell. For a rectangular parallelepiped, the Jacobian is constant all over the cell and $U_{12} = U_{12}^+$.

We explained this in the revised manuscript (page 10, line 13).

*Page 10:*
*Line 5: "four grid objects"*
*Line 13: "which should not be used for C-grids"*
*Line 16: "... describe the new objects which were added..." Line 27: "... regions, which may overlap or not... "*
*Line 28: remove "different"*
*Line 29: "... order in which they ..."*

Thank you, done.

*Page 11:*
*Line 3: "... is the velocity given in ..."*

Done.

*Page 13:*
*Line 5: "... transported through... "*
*Line 10-13: "... studies have focused on marine littler in the southern part of the North Sea (Neumann) and have included diffusion and wind drift to their model as well as used a higher resolution."*

Done.

*Line 23: "NEMO-N006" is not a standard I am familiar with, at least it does not ship with NEMO v3.6. Are you referring to the ORCA0083-N006 simulation, which is similar to ORCA0083-N001 used by Grist et al 2014 and Kjellsson and Zanna 2017?*
*Line 26: "... at horizontal resolutions of nominally 1/4 and 1/12."*
*Line 26: What is the vertical coordinate used, i.e. which interpolation scheme is used here?*

Yes, we use the ORCA0083-N006 and ORCA025-N006 models. The models use 75 $z$-layers, with the variables distributed on a C-grid. We have added those informations in the manuscript.

*Page 14:*
*Line 7: Full disclosure: this is the reason why I'm often sceptical about CMEMS data for particle modelling. All data is interpolated from the model grid to some other grid and not necessarily in a conservative way. Could you say a few words here about how this interpolation was done by CMEMS. Would you get identical results if the CMEMS*

*data came on the native C-grid from the NEMO model?*
*Line 9: "...data, which will be ..."*
*Line 10: Again, which vertical coordinate is used? Is this also interpolated by CMEMS?*

When transposing a field from a C to a A-grid, one should be particularly careful to preserve some properties, such as conservation, consistency and the boundary conditions. But even if those properties hold, information will be lost and one cannot generally obtain the same solution by interpolating the original C field and the A-transformed one, whatever being the transposing method (in CMEMS all the quantities are interpolated on the tracer grid).

So yes, we agree with you that when available, the original grid should always be used for Lagrangian modelling. However, many data are provided as a A-grid, like in CMEMS, and Parcels can also interpolate such datasets. The North West shelf reanalysis data are discretised on a $z$-level mesh, but we only use the surface surface field from this data set.

*Line 28-29: After "(Gutow 2018).": "Here we distinguish two flow types, the first based on NEMO and CMEMS data which has impermeable boundary conditions at the coast, and the second which includes Stokes drift and diffusion thus allowing beaching."*

Done

*Page 15 Line 5: "...advection." Line 6: "...are run, where if the particle beaches, it stops moving."*

Done

*Page 16:*

*Line 5: what is meant by "travelled at least once by a cell"? That the cell has been visited by at least one particle?*

You're right, this sentence is not precise. We mean that for each cell, we computed the fraction of particles that have visited it at least once. We have reformulated this sentence in the revised manuscript.

*Line 10: remove "different"*

Done

*Line 15: "... no validation of mesoscale dispersion has been done for those simulations there... ". I am fairly sure Andrew C has done some validation (AMOC strength, AABW volume etc.), but probably not for particle dispersion near the grid scale.*

By validation, we meant "validation for MP simulations". Even if the model was validated for climate quantities, this validation does not hold for particle modelling, especially at coastal scales. The sentence was reformulated to avoid any confusion.

*Line 20-22: I don't fully understand this sentence. By "differences generated in the first year" you mean "within the first year we see more transport into the Kattegat and Skaggerak leaving fewer particles for transport along the Norwegian coast"? There are some differences along the Norwegian coast and Barents Sea (less deep blue regions for Fig 6c).*

The NWS do not cover latitudes further North than 65°N, which are reached after around one year by the particles travelling to the Arctic. So basically, most of the differences between run (a) and (c) result from the first year dynamics, since further North both run do only interpolate NEMO 1/12° data. We explained that better in the revised manuscript (page 17, line 23).

*Line 27-28: "... how adding diffusion impacts the face of MP."*
*Line 28: "reduced by 68%" is in relation to NEMO 1/12?*

Yes, we precised it in the manuscript.

*Page 18:*
*I don't understand the caption of the figure. Do you mean "fraction of particles visiting each different region at least once" or "for each grid cell, fraction of particles that have visited that grid cell".?*

See previous comment about the same confusion in the main text. The caption was reformulated.

---

## Author Response (AR2)

**The Parcels v2.0 Lagrangian framework: new field interpolation schemes**

Philippe Delandmeter[1] and Erik van Sebille[1]

[1]Utrecht University, Institute for Marine and Atmospheric Research, Princetonplein 5, 3584 CC Utrecht, The Netherlands

**Correspondence:** Philippe Delandmeter (p.b.delandmeter@uu.nl)

Dear Editor,

First of all, we want to thank you for considering and accepting our paper. We are also indebted to Dr Knut-Frode Dagestad and Dr Joakim Kjellsson for their constructive comments.

Please find enclosed the sources of our manuscript "The Parcels v2.0 Lagrangian framework: new field interpolation schemes". We haven't changed anything from the accepted version except the following points:

– Parcels zenodo link: now official v2.0.0 version (we were waiting for the accepted paper. Previous link was pointing to the v2.0.0.beta version);

– Zenodo link to the scripts used in this paper. Also passed from beta to released version, even if the code didn't change since last time;

– We added an url to the NEMO data that we used, which were previously not publicly available.

Yours faithfully,
Philippe Delandmeter and Erik van Sebille